# Socio-Educational Impact of Ukraine War Murals: Jasień Railway Station Gallery

**Elżbieta Perzycka-Borowska** [1,*] **, Marta Gliniecka** [2] **, Kalina Kukiełko** [3] **and Michał Parchimowicz** [4]

1. Department of General Pedagogy Didactics and Cultural Studies, Institute of Pedagogy, University of Szczecin, 71-415 Szczecin, Poland
2. Department of Education and Childhood Studies, Institute of Pedagogy, Pomeranian University in Słupsk, 76-200 Słupsk, Poland; marta.gliniecka@apsl.edu.pl
3. Institute of Sociology, University of Szczecin, 71-017 Szczecin, Poland; kalina.kukielko@usz.edu.pl
4. Department of Pedagogy and Media Culture, Faculty of Social Sciences, Polish University Abroad, London W6 0RF, UK; michal.parchimowicz@puno.ac.uk
* Correspondence: elzbieta.perzycka-borowska@usz.edu.pl

**Abstract:** Exploring the role of public art in conveying complex socio-political messages, this article investigates the multifaceted socio-educational impact of 32 murals representing the war in Ukraine, located in *Jasień* Railway Station, Gdansk, Poland. Employing an interdisciplinary research approach, the study combines critical theory and visual communication methodologies to uncover the deeper messages conveyed by these thought-provoking murals. The analysis encompasses six diverse perspectives—historical, personal, ethical, cultural, technical, and critical—leading to the identification of six distinct mural categories: (1) resistance and hope, (2) family and courage, (3) suffering and death, (4) torturers and the oppressed, (5) animals, and (6) idyllic. The study underscores the significance of murals as a public art form for symbolically communicating social, cultural, and political events while introducing novel interpretations and expanding visual communication possibilities. Furthermore, this research demonstrates the potential of interdisciplinary approaches in exploring the intricate relationships between public art and the messages they convey, showcasing their capacity to shape public opinion and foster dialogue.

**Keywords:** murals; symbolic communication; visual communication; war in Ukraine

## 1. Introduction

In times of war, art, especially socially engaged art, becomes a tool that serves new and usually different purposes pursued by its creators than in peacetime. This is because it becomes a way of showing the truth about the battles (Rolston 2018), or an element of constructing an appropriate image in an effort to win public sympathy for political decisions taken by the state (Rolston 2018; Goalwin 2013); it is used as a form of support expressed to those caught up in armed action, to build national identity and memory (Simoes 2023; Hołda 2020). In such circumstances, artists can be witnesses, interpreters, historians of war, and those who protest most loudly against it (Kukiełko-Rogozińska 2021). An interesting expression of such activities is the use of urban space in the form of so-called street art (Pogrmić and Dercan 2021; Hoelzl and Marię 2015), in which artists, more or less openly, express their protest through murals, graffiti, paper posters, and stickers. If we treat this kind of art as a medium and urban space as a communication channel, these works become excellent material for research in the fields of visual communication, semiotics, semantics (Eco 1984; Barthes 1977; McLuhan 1967), as well as the (sub)cultural identification of authors leaving their traces in public space. Despite this scholarly potential, street art is still outside the mainstream of interest of Polish communication and media scholars (Lukaszewicz Alcaraz 2014; Perzycka 2021). Analyses undertaken in this context usually focus on its use as a tool for management and promotion of the city, as well as its impact on the city's

image as a tourist attraction. (Lachowska and Pielużek 2021). Furthermore, for many years, the dominant debate around street art in Poland was the discussion about whether it is an expression of artistic creativity or a mere act of vandalism (Moch 2016). This debate stemmed primarily from the declarations of the artists themselves, who usually expressed their opposition to various social phenomena in an illegal and destructive manner, thanks to which their works were a clear alternative to the works exhibited within the walls of galleries and museums. Today, these activities are sometimes carried out fully legally, as agreed with the city authorities and in relation to a specific public space (Kazanowski 2018).

## 2. War in Ukraine

The origins of the Russian–Ukrainian conflict can be traced back to 2000, when Vladimir Putin took office as the president of the Russian Federation for the first time (Dibb 2022). Since then, he has been trying to transform Russia into a world power again, which was to be accomplished by the construction of modern armed forces and appropriate strategies for their use, aiming to equalize the disproportions between Russia and NATO and the USA (Bergman 2023). In principle, a new-generation war waged by Russia could break out without the stage of escalating tension and be conducted in a nonlinear manner, using political, information–psychological, economic, socio-humanitarian, and, if necessary, military activities. The effectiveness of this strategy was verified in Ukraine during the Crimean operation in 2014. The main goal, which was to take over the Crimean Peninsula and join it with Russia, was achieved after the so-called referendum, using information and psychological influence (blackmail, disinformation, deception, and propaganda), combined with the activity of special forces and the army (Kasprzycki 2022a). The Russian influence in eastern Ukraine led to the escalation of separatist sentiment, which turned into an armed conflict over time. After heavy fighting in August 2014, the front stabilized, and the conflict was frozen for almost 8 years. The Russian–Ukrainian conflict turned into an open war on 24 February 2022.

The Armed Forces of Ukraine, the state structures, and the Ukrainian people put up effective and tough resistance against the Russians. Thus, the conflict exposed many weaknesses of the Russian military system and undermined the myth of a powerful and invincible army. The reasons for this state of affairs should be sought primarily in how the Russian establishment perceives Ukraine, denying its long history and recognizing today's independent state only as a kind of "anti-Russian project" of the West (Katchanovski 2022). Putin's pressure on Ukraine's pro-Russian leader, Viktor Yanukovych, not to sign an agreement with the European Union in 2013 sparked the protests that eventually led to his overthrow in February 2014. This was followed by the annexation of Crimea and the war in Donbas. After these events, Ukrainian society changed its perspective, perceiving Russia as a source of threat to its integrity and sovereignty. As a result of these developments, Russia's political influence in Ukraine began to decline, and Ukraine's increasingly stronger efforts to integrate with Western Europe deepened the Kremlin's frustration. Fighting for Ukraine and their own sphere of influence, the Russians decided to use force (Kasprzycki 2022b). The Russian aggression against Ukraine in 2022 was of course a shock to the international community, but it was not a surprise (US and NATO intelligence sources warned about it). What turned out to be more surprising was its course and, above all, the effectiveness of Ukraine's resistance in the confrontation with Russia. Despite Russian aggression, the Ukrainian state has not lost its sovereignty and ability to manage the territory it controls (Bendyk and Buras 2022). Using well-thought-out tactics, the Ukrainian forces were able to slow down and even stop the Russian offensive quite effectively. Russian logistical problems also deepened at a rapid pace. The West began introducing new sanctions, and Ukraine was flooded with weapons from the USA, Great Britain, Poland, and other countries. In addition, Ukraine received valuable intelligence data from NATO countries and trained its soldiers on their soil to operate new equipment (Kasprzycki 2022a).

### 3. The Social and Educational Importance of Murals

The involvement of artists in creating murals dedicated to the war in Ukraine is an excellent example of the importance of this form of communication in creating a narrative about issues important to the international community. In this context, murals are important for raising awareness about the Russian–Ukrainian conflict and its impact on people's individual and social lives. They depict the struggles, challenges, and emotions of those directly or indirectly affected by this war. Creating anti-war and pro-peace murals is a phenomenon that we can currently observe not only in Ukraine and neighboring countries but also worldwide. As a public form of art, murals offer a unique opportunity to communicate content that has personal, informative, and emotional potential, as well as social potential that can be appreciated by a wide audience. They have the power to fill gaps in the understanding of the political, social, and cultural aspects of war, enabling a fuller understanding of the experiences of people coping with the effects of war (Gralińska-Toborek 2019).

The educational potential of murals is also important, especially in the context of visual education. In principle, they are carriers of images whose task is to inspire thinking and experiencing (Pasieczny 2016; Corbisiero-Drakos et al. 2021). They can also be treated as a tool enabling audiences of different ages and cultural capital to become acquainted with historical events in an accessible and engaging way. This is especially important in educating young people particularly in the context of war and its consequences. This form of communication is conducive to perceiving and understanding what is challenging to assimilate within school walls (Mayer 2016). Based on the conscious, empathic involvement of students, we can successfully use murals to work on open thinking about socially critical phenomena (Stano and Żądło 2021) and, thus, teach a critical approach to messages from the media and other sources (Perzycka and Łukaszewicz-Alcaraz 2020).

In the article, we discuss an example of a series of murals that are used to express protest against the war in Ukraine and provide emotional support for the citizens of this country in their fight for independence. The subject of our reflections is the project "Solidarity with Ukraine", carried out in Gdansk (Poland) as an expression of the artists' opposition to the Russian–Ukrainian war and its consequences. Our analyses are based on Paul Martin Lester's (2011) concept of visual communication perspectives: personal, historical, aesthetic, cultural, ethical, and critical.

### 4. Research Methodology

The war in Ukraine resulted in the creation of murals in many Polish cities, which were an expression of opposition to the Russian attack. An example of such activities is the gallery of murals "Solidarity with Ukraine" created on a 300 m wall by the railway embankment in Gdansk. The project coordinator is Prof. Adam Chmielowiec, supervised artistically by Prof. Jacek Zdybel. The authors of the works in the gallery are mostly students of the Academy of Fine Arts in Gdansk and the Faculty of Arts of the Maria Skłodowska-Curie University in Lublin, as well as individual artists who represent various cities (e.g., Gdansk, Wroclaw, and Lublin) and various nationalities (e.g., Polish, Ukrainian, and Belarusian). Works have been appearing regularly since March 2022, and more artists can still submit their ideas. The self-government of the Pomeranian Voivodeship, Polish Metropolitan Railway SA, and the Academy of Fine Arts in Gdansk took the patronage over the initiative. The patrons did not impose restrictions on the artists; the only remarks were that they should create works with sociopolitical overtones, avoiding the language of hatred (Umięcka 2022).

The perception of murals in public spaces usually occurs in a state of distraction. As part of our research and an effort to objectify the analysis, we took photos of the murals from the "Solidarity with Ukraine" gallery. Photographs of murals allow us to see works that are difficult to access or even invisible in physical space. This form of reception is, of course, different from experiencing their physical presence (scale, colors, location, and interaction with the environment can disappear in the photographs), but allows for wide sharing and dissemination of their content (Gralińska-Toborek 2019). We assume that these

photographs are visual messages, the reading of which, depending on the context of their location (cultural, religious, political, or economic), is subject to the dynamics of change. Therefore, we refer to the methods of visual theory (Konecki 2022; Bryant and Kathy 2007; Engeldrum 2004; Freedberg 2021) and visual communication (Lester 2014). The adopted assumptions allowed us to recognize the interdependence of people (creators and audience) and their culturally defined messages. The theory of visual communication, in turn, made it possible to apply a research approach to photography, understood as a visible message in all these accounts by extending the definition of the image beyond its place in the history of art (Freedberg 2021). On the basis of Lester's (2011) proposal, we focus our research and the resulting explanations on six perspectives of visual communication: (1) personal perspective—the viewer has an opinion about the image based on their own understanding and cultural influences; (2) historical perspective—considering factors such as it historical context, its creation, the events and social dynamics that influenced it, and how it may have impacted the thinking and actions of its time; (3) aesthetic perspective—analyzing the aesthetic values and perception of the image, including elements like composition, color, and style, (4) cultural perspective—recognizing the identity of the image referring to its content, dynamics, and symbolism, (5) ethical perspective—considering the ethical issues, the portrayal of individuals or groups, and any potential biases or manipulations, (6) critical perspective—evaluating the image through a critical lens, questioning the underlying it might convey, and reflecting on how it may contribute to, challenge existing discourses. To effectively analyze the perception of murals in public space, it is imperative to adopt the fifth (ethical) and sixth (critical) perspectives as an analytical framework that can be applied to all defined perspectives. Such an approach is necessary to create a uniform, coherent narrative about the gallery of murals in Gdansk, which is shaped by individual researchers' subjective points of view regarding unique murals.

Thanks to the ethical perspective, we have become responsible for not separating the creator, the recipient, and the image in the analyses. Using a critical perspective, we could look at the murals more systematically to make the perception more objective. We looked at the murals from two perspectives simultaneously: individual and collective. This approach allows for an inclusive and diverse analysis that considers the different experiences of researchers and, at the same time, enables the creation of a more complex and meaningful narrative about anti-war murals. Our approach is illustrated in Figure 1 in the context of Lester's perspectives.

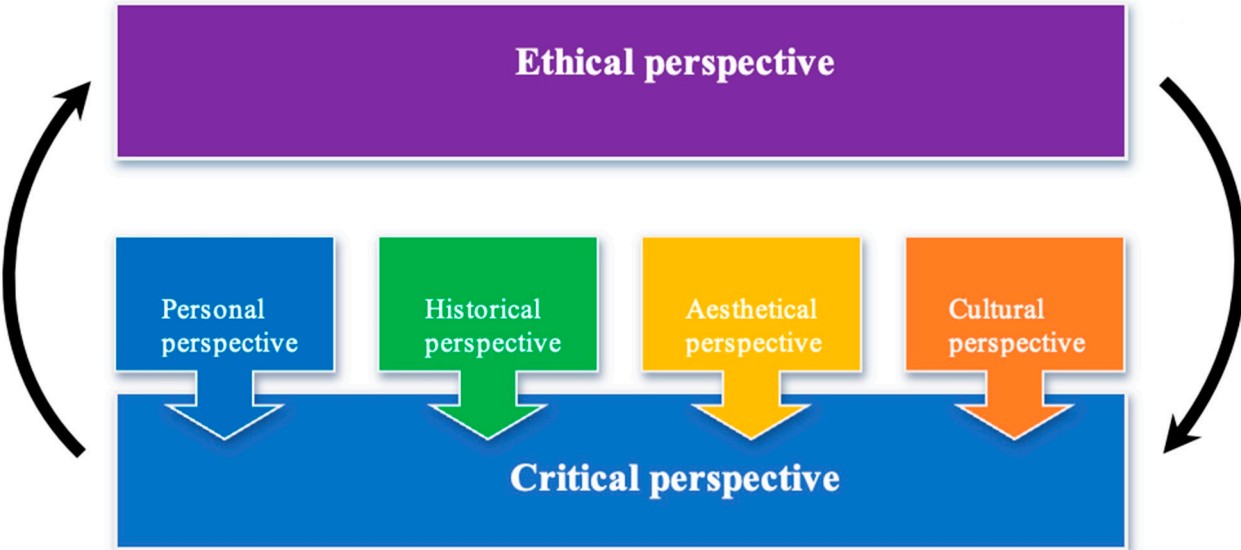

**Figure 1.** Interdependence of six perspectives of visual messages after P.M. Lester. Source: Own elaboration based on P.M. Lester (2011).

To conduct the analysis, the researchers personally took photos of the murals on 12 January 2023. In total, 32 photos of unique murals, along with four contextual images of the surrounding environment: the railway embankment, the railway station (*Jasień*) in Gdansk, and two images forming the horizontal layout of the gallery. Additional photographs were taken to capture the visual record of the murals in the context of analysis and interpretation in the research process.

## 5. Analysis of Murals as Visual Messages

The discussed murals are works of art placed in public spaces, thanks to which they are available to a broad audience. The main goal of the "Solidarity with Ukraine" gallery is to convey the message of condemnation of war crimes and to provoke viewers to reflect and discuss this war, as well as war in general (Umięcka 2022). The murals vary in style and meaning, but they share an anti-war message and a desire to express support for Ukraine and its inhabitants.

The analyzed project is, therefore, an expression of opposition to the war and its consequences. It was not without reason that it was created in Gdansk. The city is the cradle of the "Solidarity" movement, which operated in opposition to the communist regime in the 1980s, playing a vital role in the fight for freedom and democracy in Poland. It can be said that Poles, due to their experience of war and occupation, are particularly sensitive to anti-war topics.

The murals were created in a hard-to-reach location that is dangerous due to the trains passing close by. As a result, their audience primarily involves people who travel by train, run, and go for a walk in the forest. You can look at the gallery through the prism of individual paintings in their series, observing stories about the cruelty of war, lost hope, and the fight for dignity (Figure 2).

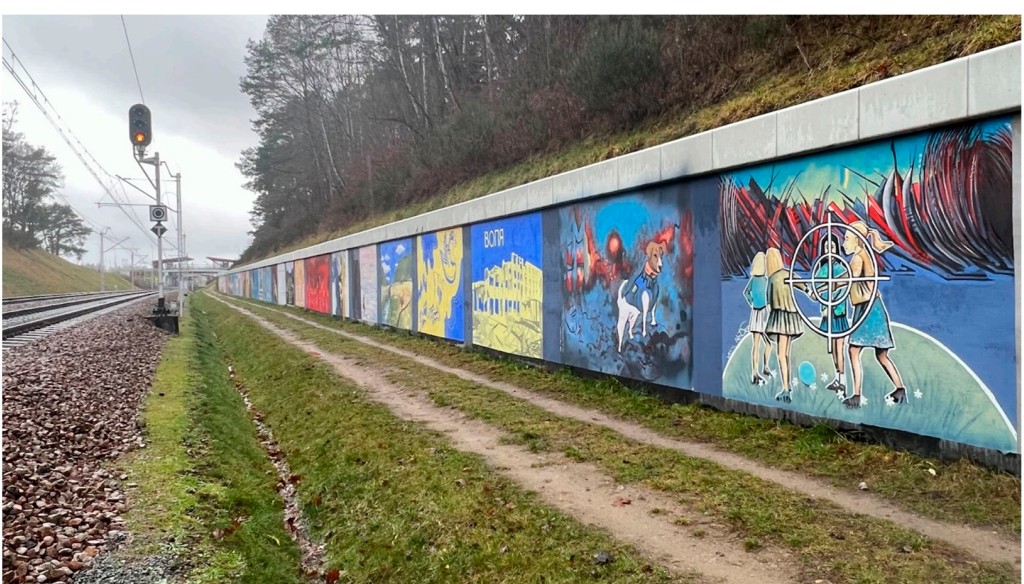

**Figure 2.** Systematics of the murals.

Murals at the PKM Gdansk *Jasień* station show the dramatic effects of the war in the form of ruins of cities, destroyed residential buildings, and landscapes depicting natural spaces devastated by the enemy. In addition, images of violence appear on the murals both in a realistic context (a mural commemorating the tragic war crime in Bucza, in March 2022, by the Land Forces of the Russian Federation during Russia's invasion of Ukraine) and in an abstract dimension (skeletons in Russian helmets with the letter "z"—reminiscent of the danse macabre theme; a field of the sunflowers under attack from falling bombs).

## 6. Interpretation of Symbolism in Murals

In the realm of murals, symbolism plays a significant role in conveying meaningful messages that evoke thought, emotion, and reflection. According to P.M. Lester, the interpretation of visual communication is subjective, depending on the person analyzing the visible message. The symbols used by the artist's murals, however, allow not only for their more unambiguous interpretation but also for embedding them in the broader historical, social, cultural, and political context in which they were created.

We gathered that the choice of symbolism is not accidental because the creators of the murals are artists with extensive social awareness and experience in the field of visual arts (Works by the artist Piotr Jaworski "Tsue" and students of the Academy of Fine Arts in Gdańsk under the supervision of professors of the university: Jacek Zdybla, Maciej Świeszewski, Krzysztof Polkowski and Mariusz Waras). As a result of recognizing the themes, motifs, and symbolism, we identified six categories of murals, each of which has unique characteristics and meaning. We briefly introduce these categories, examples of murals, and their symbolic meaning, which we defined by applying the lens of knowledge of our scientific disciplines: art, cultural studies, sociology, and pedagogy.

## 7. Murals of Resistance and Hope

Across the globe, murals pulsate with expressions of defiance and aspirations for a brighter future. They draw upon an eloquent visual vocabulary, employing an assortment of images and symbols to construct potent narratives that convey social and political commentary while also celebrating the resilience and spirit of communities (Young 2014).

In the heart of Washington, USA, for instance, the monumental "Black Lives Matter" mural reverberates with the collective outcry sparked by the tragic killing of George Floyd (Asmelash 2020). This expansive piece, brought into existence by an alliance of artists and activists, has transformed an urban thoroughfare into a striking protest against police violence, systemic racism, and the consistent marginalization of Black communities within the United States. As a symbol of unity and resistance, the mural stimulates dialogue and mobilizes activism, fostering a palpable sense of solidarity among advocates of racial justice and reform.

Across the Atlantic, in Belfast, Northern Ireland, the Women's Mural stands as a stirring tribute to remarkable women who bravely championed equality and justice (Rolston 1991). This compelling visual narrative not only commemorates their invaluable contributions to society but also serves as a beacon of hope and inspiration, encouraging the ambitions of future generations (Young 2014).

Moving further east, murals portraying resistance and hope are a central theme in the "Solidarity with Ukraine" collection. Here, the artistry employs a poignant metaphorical juxtaposition between sunflowers and military weapons (Figures 3–6).

The sunflower, a culturally significant emblem for Ukrainians, has been a longstanding symbol of national identity and defiance, particularly during the Soviet era. Throughout the 1970s and 1980s, Ukrainian dissidents and political prisoners sported sunflower patches as a tacit show of solidarity in their quest for independence and opposition to Soviet subjugation (Shkandrij 2009). This peaceful emblem, contrasted against the harsh realities of warfare (www.smithsonianmag.com, accessed on 17 April 2023), communicates the tenacity of Ukrainian resistance and their hopes for triumphant victory. The alliance of the sunflower symbol with Ukrainian identity and defiance might illuminate why these murals recurrently feature explicit military equipment.

Dominating the mural, the T-90 tank poses a menacing manifestation of conflict, epitomizing the destructive forces poised to annihilate life and liberty. It embodies the suffocating authority that relentlessly yearns for dominance and control.

In juxtaposition, the sunflowers entwining the monolith of the tank project an emblem of life, optimism, and renewal. These vivid blossoms, firm upon their stalwart stems and extending leaves, bear witness to the tenacity of existence and the resilient spirit that ceaselessly strives to surmount adversities.

Additionally, the artist's choice of a monochromatic spectrum in the mural instills a deeper stratum of symbolism into the tableau. The stark polarity between black and white serves to underline the ruthless realities of warfare and the unmistakable schism between the forces of benevolence and malevolence, of existence and demise, and of hope and desolation. The austere black-and-white palette intensifies the gravity of the struggle while insinuating the impossibility of neutrality within such a tumultuous confrontation.

Paulina Sosińska's mural unfurls a compelling and introspective panorama that expertly juxtaposes the vibrant allure of nature against the menacing specter of violence and annihilation. The sunflower field, resplendent with robust, oversized blossoms, stands as a symbol of life, optimism, and vivacity. These steadfast flowers, unyielding in their stature, confront their nebulous future with audacious courage and resolution, embodying the tenacious spirit of the innocent ensnared in the maelstrom of war.

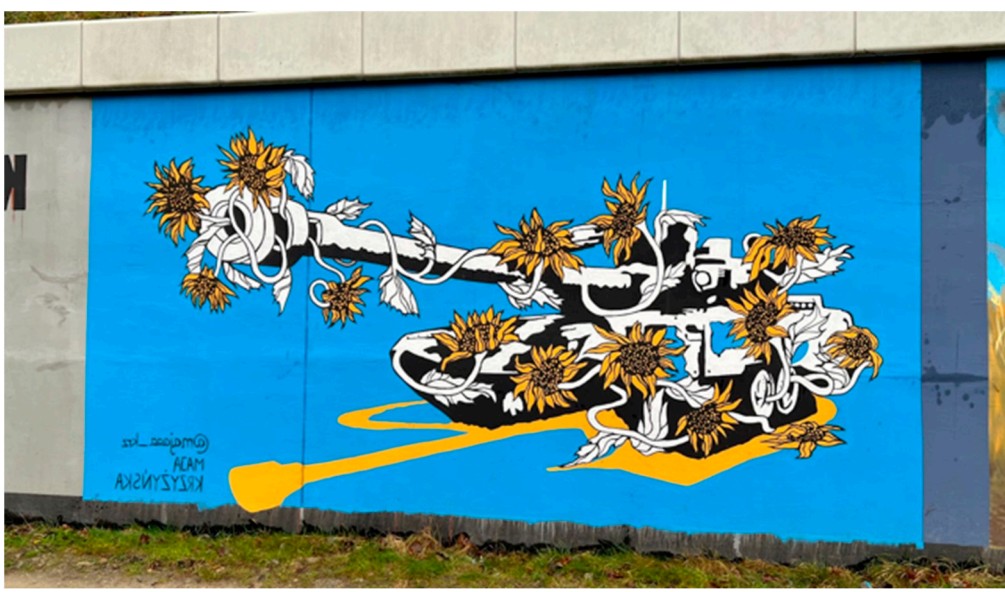

**Figure 3.** Artist: Maja Krzyżyńska.

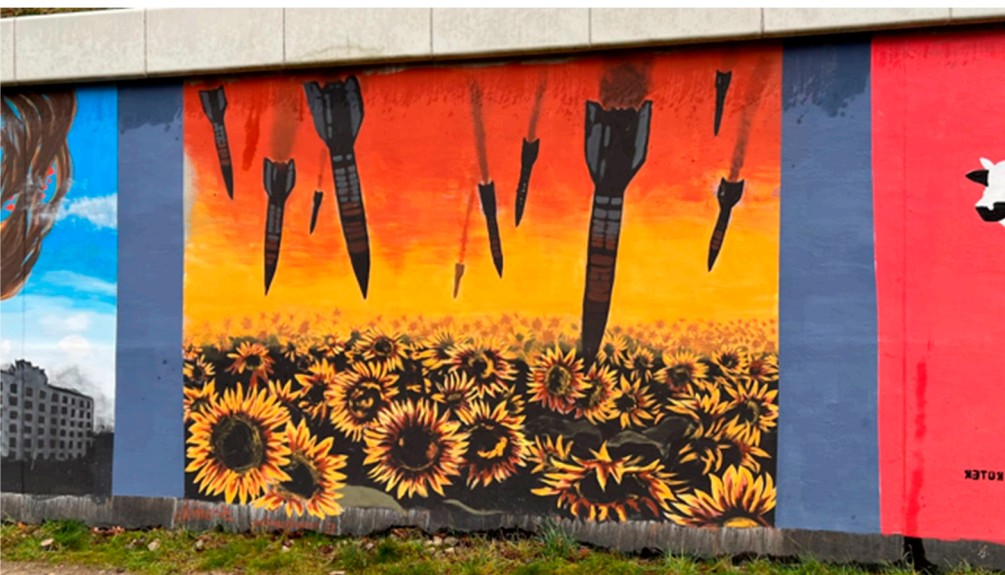

**Figure 4.** Artist: Paulina Sosińska.

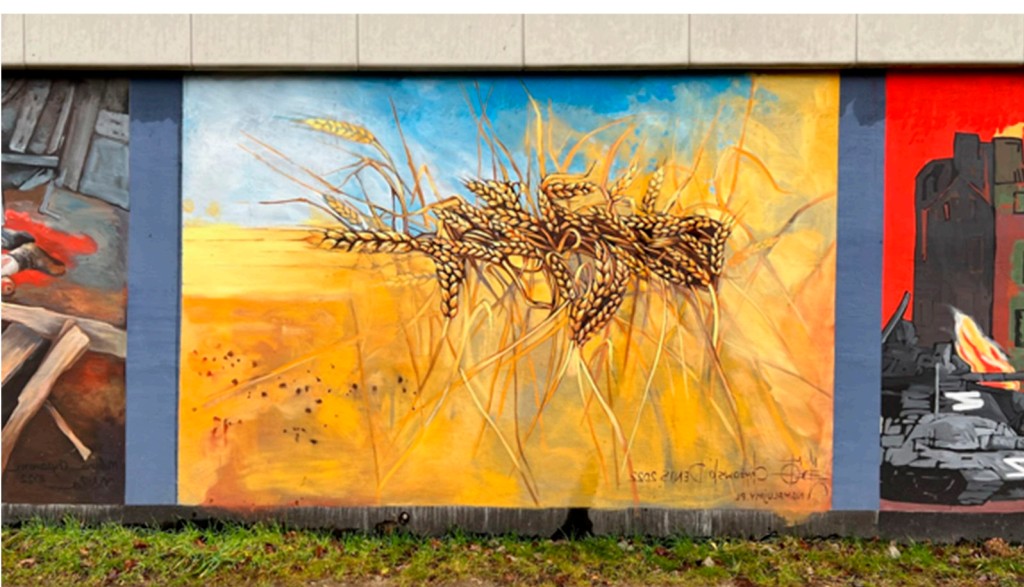

**Figure 5.** Artist: Denis Chyżawski.

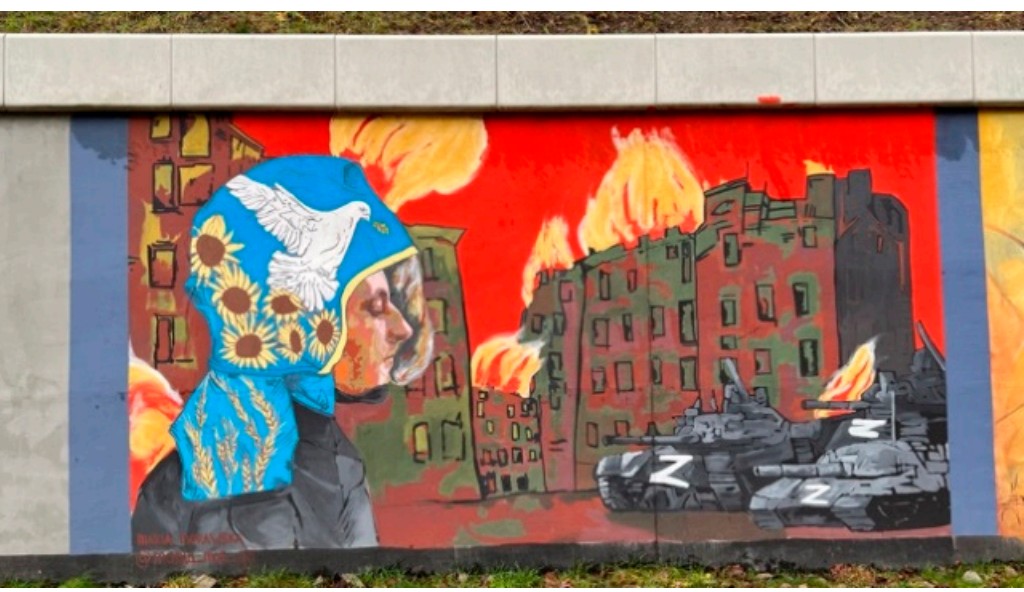

**Figure 6.** Artist: Maria Morawska.

The depiction of bombs raining down from the sky introduces a palpable undercurrent of tension and anticipatory peril, underscoring the fragility of existence and the perpetual menace haunting those dwelling in conflict-ravaged territories. The bombs serve as a chilling memento of how swiftly the serenity and splendor of the natural world can shatter, punctuating the cataclysmic ramifications of human discord.

The backdrop of the mural, painted in arresting shades of red and yellow, imparts a sense of pressing urgency and tumult. These fervent and intense colors can be interpreted as metaphoric representations of the passions and emotions fueling human strife, while also encapsulating the devastation and disorder that war indiscriminately wreaks upon landscapes. As the sunflowers stand undeterred in the face of plummeting bombs, the mural underscores the immense power of resilience and hope amidst adversity. Despite the looming danger, the sunflowers persist in their growth and thriving existence, embodying the indomitable human spirit's determination to persevere and flourish even under the direst circumstances.

Natural landscapes often harbor an embedded message, unearthing profound insights beneath their apparent tranquility. In the composition of Denis Chyżawski, this notion resonates through his ingenious arrangement of wheat ears fashioned into the likeness of a rifle—an eloquent testament to the resilience and steadfastness of Ukrainian soldiers.

In sculpting the wheat ears into the form of a firearm, the artist orchestrates an intriguing fusion of fertility, sustenance, and defense, crafting a metaphor replete with layers that reverberate with the endurance and fortitude of Ukrainian warriors confronting adversity. The wheat ears stand as a beacon of life and nourishment, underlining the vital role of agriculture in nourishing communities and securing their survival. This symbol concurrently signifies the deep-seated bond between the populace and their fertile homeland, highlighting the integral role it plays in shaping their identity.

The metamorphosis of these nourishing wheat ears into a rifle encapsulates the inherent potency within nature and its dual capacity to nurture and safeguard. This potent visual metaphor encapsulates the resolute spirit of the Ukrainian soldiers, drawing their inexhaustible strength and determination from the very soil they valiantly defend. Furthermore, the blending of natural elements and weaponry underscores the human existence's poignant duality—the innate potential for both creation and obliteration.

The artist's deliberate choice to amalgamate these ostensibly contradictory elements mirrors the intricacies of warfare and the perpetual tug-of-war between life and death, or between tranquility and violence. By harmoniously merging nature's elements with the imagery of weaponry, the artist adeptly portrays the paradox inherent in human conflict and the extraordinary capacity for resistance and perseverance discovered even within the bleakest circumstances.

The mural by Maria Morawska presents a vivid portrayal of everyday heroism and resilience in the face of adversity. The firefighter, a central figure in the mural, embodies the strength, courage, and dedication of individuals who confront danger and disaster in the pursuit of protecting their country and its people. The painting is also a tribute to the dedication and heroism of the rescue services. The dual challenges faced by the firefighter, battling both the raging flames and the hostile forces represented by Russian tanks, convey a metaphorical narrative of resistance and determination. The firefighter's struggle against these two powerful elements highlights the immense physical and emotional challenges faced by those defending their homeland and fighting for freedom. The inclusion of national symbols of Ukraine, such as the sunflowers and the image of a pigeon, woven into the firefighter's blue uniform, emphasizes the strong connection between the individual and their nation. The sunflowers signify hope and perseverance, while the pigeon serves as a symbol of peace and unity. These elements not only strengthen the firefighter's resolve but also emphasize the core values that inspire him to continue his fight against the destructive forces at play. In paying tribute to the heroism of the rescue services, the mural also acknowledges the sacrifices and commitment of countless individuals working tirelessly to maintain safety and order amidst the chaos of conflict. The firefighter's courage and determination in the face of seemingly insurmountable odds serve as a powerful testament to the strength of the human spirit and the unyielding pursuit of freedom and justice.

## 8. Murals about Family and Courage

War, with its merciless tide, cleaves hearts, compelling individuals to abandon their homes, venture into the battlefield, or embark on an odyssey of survival. The mural vividly encapsulates this poignant reality, as the characters, despite their close ties, grapple with the uncertainty of survival and the prospects of reaching their aspired destination. What does destiny hold for them as they tread toward the end of this arduous journey—their unseen future?

The exploration of familial bonds in the crucible of war deeply resonates with the audience, leaving an indelible imprint on their consciousness. Natalia Świerczyńska's mural serves as a powerful testament to this theme (Figure 7). It narrates the silent tale of an adult, guiding a child by hand, navigating the treacherous landscapes of conflict.

The symbolism of the rifle slung over the adult's shoulder juxtaposed against the innocent burden of a backpack on the child crafts a narrative ripe with profound meanings. It stands as a stark reminder of the cruel dichotomy of war—where symbols of protection and innocence intermingle, capturing the essence of human resilience and the desperate hope for a safer future.

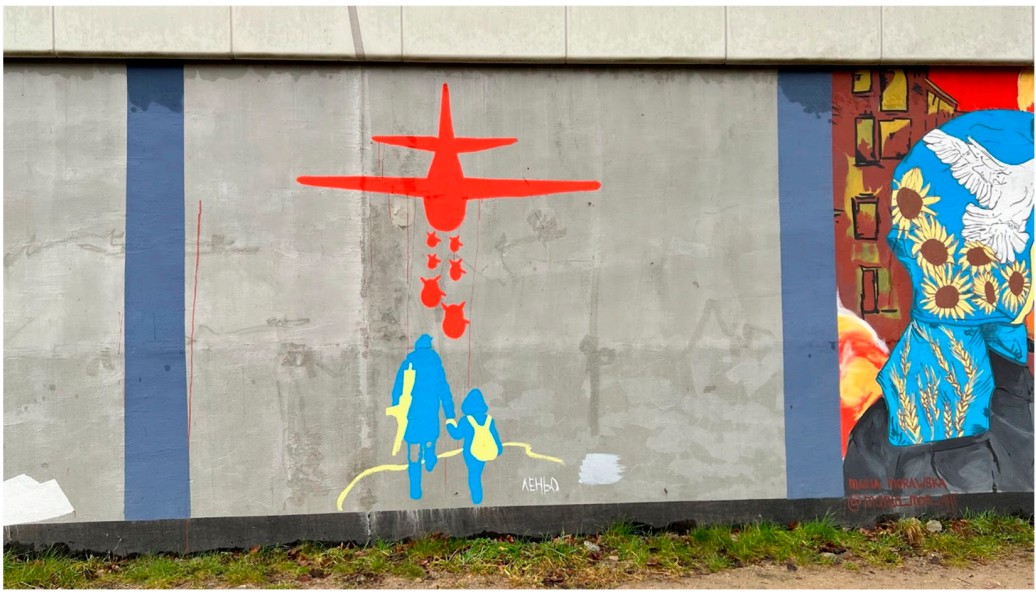

**Figure 7.** Artist: Natalia Świerczyńska.

The stark blues and vibrant yellows, representative of the figures and the symbols—the rifle and the backpack—echo the colors of the Ukrainian flag, imparting a patriotic undertone to the scene. The rifle, resting on the adult's shoulder, bears the weight of violent conflict and the imperative duty of safeguarding family amidst turmoil. It reflects the intricate decisions individuals are coerced into, compelling them to wield weapons to safeguard their kin and homeland.

The sight of the rifle, a grim reminder of the personal sacrifices endured by countless individuals during the war, unveils the harsh realities confronted by those thrust into combat for the sake of their progeny. Simultaneously, the backpack signifies the premature forfeiture of innocence and the heavy onus of responsibility thrust upon young shoulders during warfare. Its presence underscores the sobering fact that even children cannot escape the reverberations of armed conflicts, often necessitating them to shoulder responsibilities surpassing their tender years. This baggage also signifies the disruption of normal childhood experiences, as children might be compelled to abandon their homes, schools, and companions to evade war's imminent dangers.

These potent symbols reverberate with viewers, stirring empathy and fostering a deeper comprehension of the impact of armed conflict on the familial fabric. Vivid imagery encapsulates not just the tangible manifestations of war but also conveys the intricate emotional struggles experienced by individuals directly afflicted by conflict.

By integrating these elements into the mural, the artist beckons viewers to contemplate the personal sacrifices and hardships endured by those confronting the devastating ramifications of warfare. Observers are encouraged to engage in an emotional reciprocity process, wherein they envision themselves in the depicted individuals' predicaments. This empathetic connection enables them to acknowledge the adversities faced by families in conflict-ridden zones, such as loss, grief, displacement, and the daily struggle for survival. The stirring of empathy serves as a potent catalyst for change, nurturing broader awareness, understanding, and compassion for families caught in war's merciless crossfire.

It is pertinent to highlight that the mural finds inspiration from a globally circulated photograph taken at the war's onset, symbolizing the dedication and bravery of Ukrainians demonstrating exceptional courage.

Real events frequently inspire mural creators (Figure 8). A man and a woman, donned in military attire, share a tender kiss. The body language of these characters exudes affection and commitment, surviving the hardships they face. Moreover, the military uniforms signify that the bond between the couple was tempered in the crucible of war.

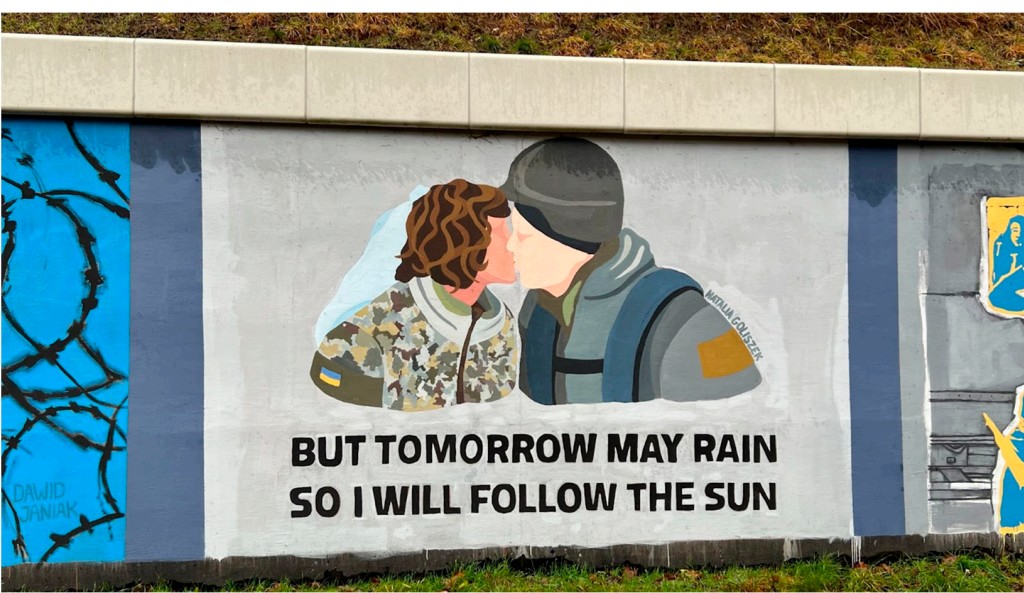

**Figure 8.** Artist: Natalia Goliszek.

The mural vividly captures the wedding of Lesya Ivashchenko and Valeriy Filimonov, courageous members of the Ukrainian Defense Forces who, despite the war's shadow looming over Kyiv, decided to solidify their bond in matrimony (Umięcka 2022). An English inscription enhances the mural's essence: "But tomorrow may rain so I will follow the sun." This lyrical excerpt, seemingly borrowed from the Beatles' song "I'll Follow the Sun" from their 1964 album "Beatles for Sale", primarily penned by Paul McCartney, imbues the mural with symbolic significance. It embodies the couple's steadfast resolve to seek out rays of hope and moments of happiness amid the foreboding clouds of uncertainty and adversity.

In the face of daunting trials, the couple actualized their dreams, demonstrating the power of love and resilience to surpass fear and danger. The mural radiates a potent message of hope, underlining love as a force of such monumental strength that it dwarfs the fear of death. Moreover, it extols the virtues of perseverance and the unwavering commitment to uphold core values, even when tragic circumstances conspire to erode them.

Through their personal narrative, Lesya Ivashchenko and Valeriy Filimonov offer a beacon of inspiration, urging others to hold fast and contend for what they hold dear, regardless of the daunting challenges life may present. They exemplify the indomitable spirit that enables us to weather life's storms, standing as a testament to the enduring power of love and hope even amidst the most turbulent of times.

## 9. Murals about Suffering and Death

In *Jasień*, one encounters murals imbued with potent images, serving as visual metaphors for suffering and death. Certain murals tap into a rich tapestry of universal symbols, creating a dialogue that resonates across cultural boundaries. Prominent among these is the timeless motif of the "dance of death" (danse macabre) (Holbein 2016), depicted alongside faceless visages and disembodied hands. Rooted deep within the cultural psyche,

skulls and skeletons form a recurring theme, emblematic of life's fleeting nature and the inevitability of death that remains an unflinching constant (Figures 9 and 10).

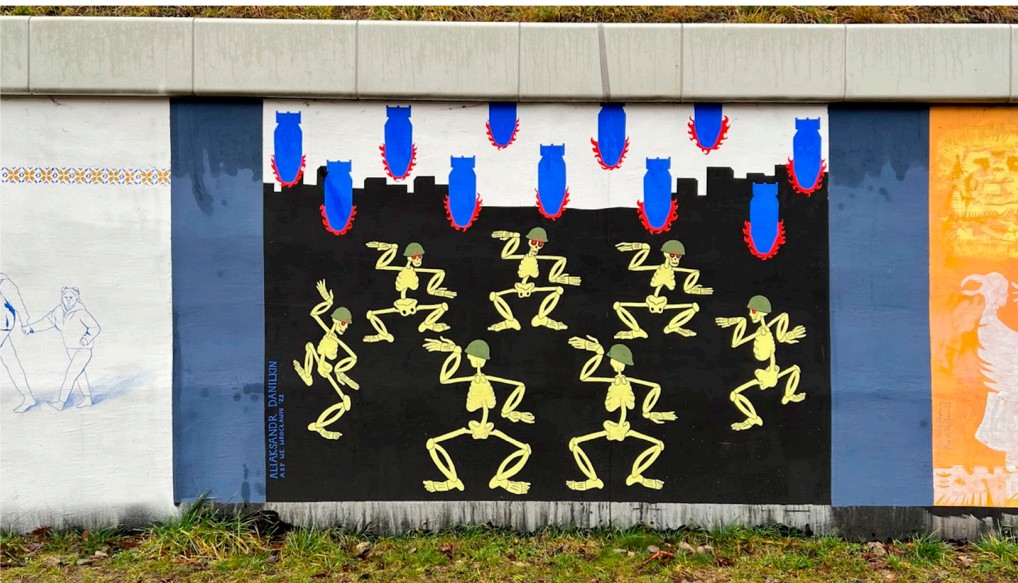

**Figure 9.** Artist: Aliaksandr Danilkin.

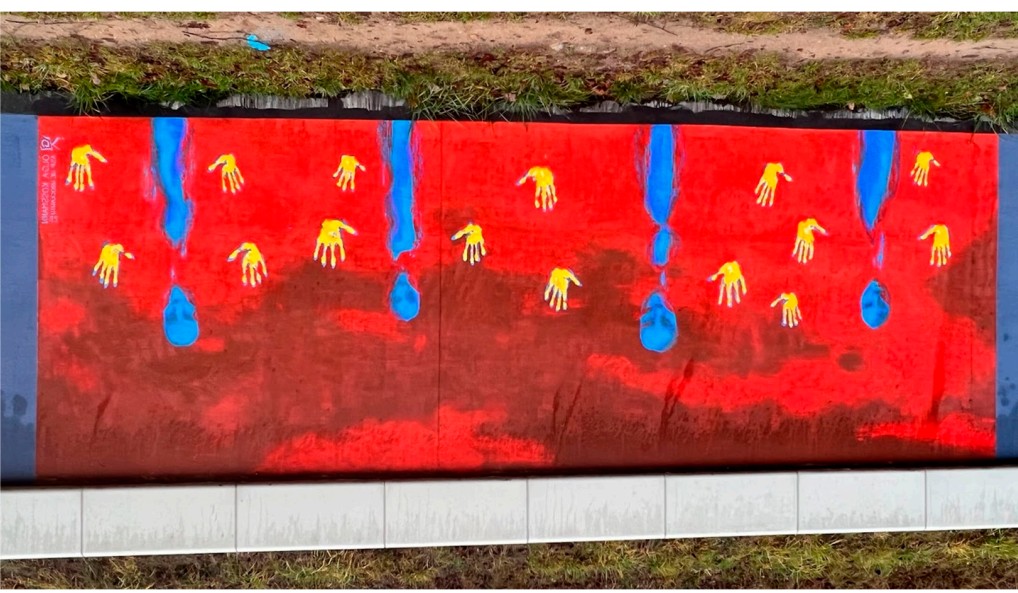

**Figure 10.** Artist: Olga Kossman.

The "dance of death" symbolically captures humanity's inexorable march toward mortality, a dance with destiny where everyone—regardless of wealth or poverty, and of power or impotence—must participate, united by the shared destiny of mortality. This potent visual narrative underscores the delicate balance and ephemeral nature of life, while starkly reminding us of death's indiscriminate reach.

The artist's depiction of faceless characters and disassociated hands eloquently expresses a poignant sense of despair, powerlessness, and alienation in the face of pervasive suffering and inevitable death. These ethereal images echo feelings of profound solitude and detachment, hinting at the dissolution of identity and individuality that often shadow the experience of war and widespread devastation.

Skulls and skeletons in the murals serve as enduring symbols of transience, amplifying the unforgiving inevitability of death that shows no favoritism (Figures 9 and 10).

These quintessential symbols, charged with existential significance, serve as potent reminders of our transient existence and the ultimate futility of power struggles, greed, and territorial conquest.

Through the articulate usage of powerful visual metaphors, murals challenge viewers to confront the unsettling realities of suffering and death, triggering profound introspection about the human condition and the profound consequences of war. Leveraging universally recognized symbols, the artist transcends cultural and linguistic barriers, crafting an evocative narrative that resonates with the shared human experience, inducing a sense of collective empathy and understanding.

In the context of war, the mural "Dance of Death" by Aliaksandr Danilkin takes on a particularly poignant meaning, emphasizing the fragility of human life and the devastating effects of conflict. The allegory is a powerful reminder of the human victims of conflict, highlighting lives lost and communities devastated by violence. Furthermore, the allegory of the "dance of death" in the context of war can be seen as a critique of the political forces that drive the conflict and its toll on human life. In the Danilkin's mural, the pose of the skeletons wearing Russian helmets is arranged in the form of the letter "z", which refers to the swastika and the ideology of Nazism. In turn, attacking fighters in blue are a symbol of Ukraine and constant resistance to the enemy.

Another example is the abstract mural by Adam Chmielowiec and Tomasz Wiktor (Figure 11). In the central part of the mural, the lower part of the mural shows embroidery, a symbol of Ukrainian folklore. In the mural, two women are in folk costumes. The author presents the same scene from two different perspectives: before the war's outbreak and after the conflict started. The women are standing under a branchy tree with the sunflower floating above it. The woman in peacetime stands surrounded by birds and holds a musical instrument in her hands. The mural has a cheerful meaning, but the unexpected outbreak of war breaks the peace of the idyllic landscape. On the right side, birds are taking off, a symbol of migration, and a woman is holding a rifle against the background of approaching fighter planes. In the central part of the painting, there is a blooming tree, which can symbolize constant development but also alternative outcomes and choices in life. The war disturbed the peace and left a lasting impact on the lives of ordinary people. Falling buds can symbolize the loss of life; there is smoke from an explosion, which may represent a symbolic beginning of the war. Next to it—on the left and right—artistic smears of paint flow down the mural like tears and blood. The overwhelming gray of the explosion is juxtaposed with the national colors of Ukraine, thanks to which the message gains symbolic meaning. The authors suggest that the unexpected aggression caused much pain and suffering, and the armed conflict deprived Ukraine of its sense of security and sovereignty. War leaves a permanent mark on social consciousness, generating several conflicts. The mural is a clear sign of protest against all forms of violence.

The mural by Iwona Sidło (Figure 12) with the inscription "No Signal" is also very telling. War is also an attack on freedom of speech. This mural is a multilayered metaphorical commentary on the consequences of war and the vital importance of communication. At a physical level, the mural highlights the devastating impacts of war on infrastructure, such as the destruction of electrical systems and the resulting blackout of entire cities. This representation serves as a stark reminder of the tangible and immediate costs of war, emphasizing the disruptions to everyday life and the broader implications for the communities affected. Beyond the literal, the mural also delves into the deeper issue of communication breakdowns between nations and ideologies. The "No Signal" inscription metaphorically alludes to the lack of dialogue and understanding between the East and the West. This division extends not only to geopolitics but also to the restricted flow of information within countries such as Russia and Belarus, where freedom of speech is often suppressed or controlled. By addressing the attack on freedom of speech, the mural calls attention to the danger of censorship and the critical role that open communication plays in fostering understanding and unity. War, as a weapon against free speech, stifles dissenting voices and inhibits the sharing of ideas and perspectives, further perpetuating conflict

and division. By juxtaposing the physical devastation with the consequences for freedom of speech and the flow of information, the mural serves as a powerful reminder of the importance of communication, empathy, and understanding in overcoming the challenges posed by conflict.

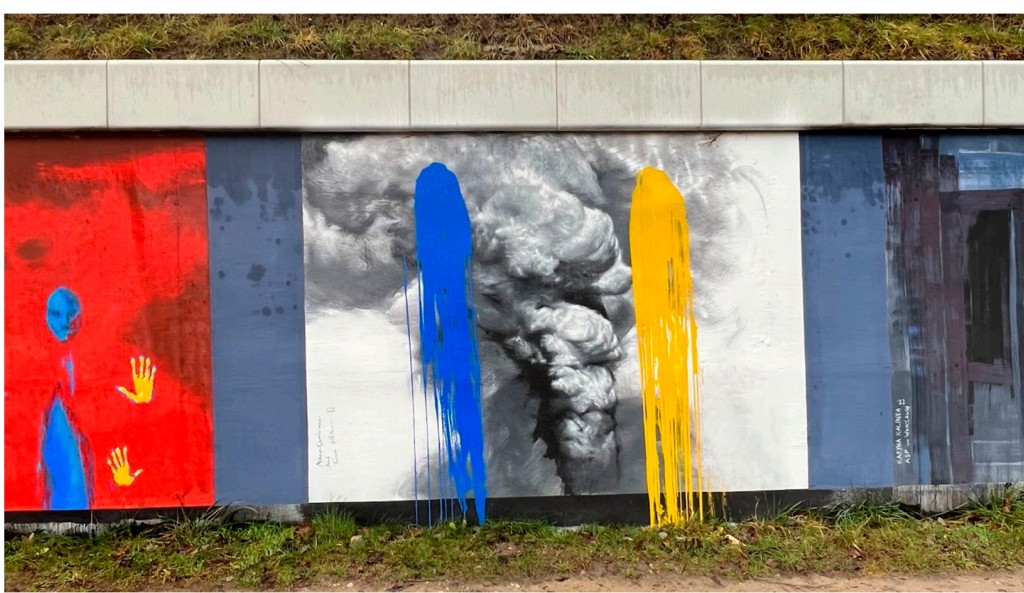

**Figure 11.** Artists: Adam Chmielowiec, Tomasz Wiktor.

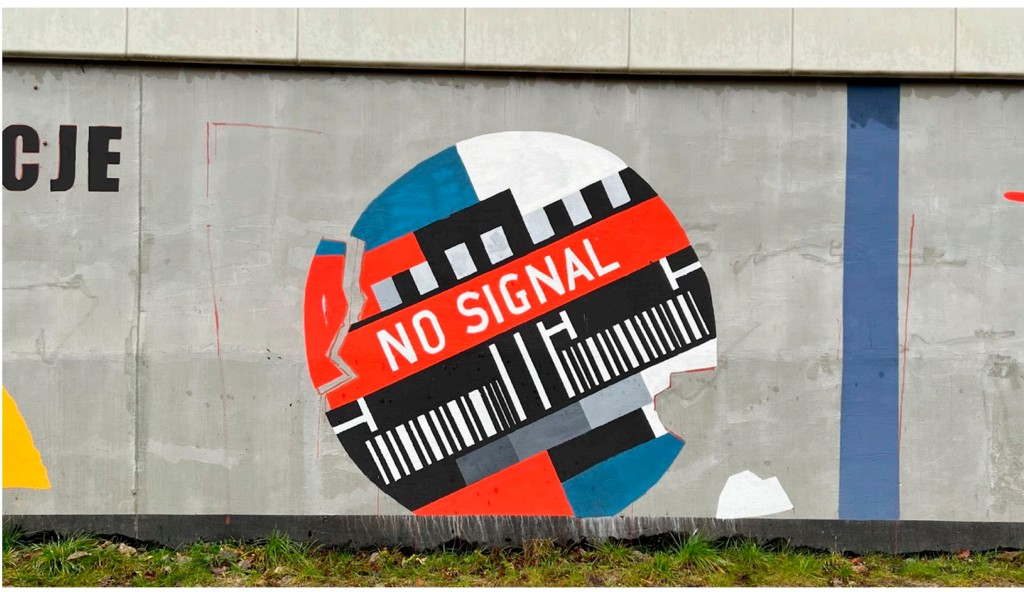

**Figure 12.** Artist: Iwona Sidło.

Murals that depict suffering and death can be harnessed as poignant memorials, honoring the fallen soldiers, civilians, and other casualties of war. The artists, in their pursuit of verity, deliberately employ potent symbolism and heart-stirring emotional resonance to pay homage to those who laid down their lives in defense of their homeland's freedom. These symbols, designed to arrest attention and etch a lasting imprint in the observer's memory, transform these murals into living tributes.

These art pieces project the raw emotions and stark reality of war, thereby fostering a greater consciousness of the aftermath of armed conflicts. They incite conversations on the tenets of peace, the nuances of diplomacy, and the complex dynamics of international

relations. These murals, while acknowledging the harshness of war, serve as powerful advocates for harmony, inspiring a dialogue that resonates beyond borders. They are not just silent witnesses to the past, but also potent tools in shaping discourse about the present and future, promoting a more peaceful and empathetic world.

## 10. Murals about Torturers and the Oppressed

The mural opening the gallery (Figure 13) was created by the artists from Gdansk, Piotr "Tuse" Jaworski and Mateusz Rybka. The artists have created a powerful narrative that examines the complex interplay of heroism, power, and the human cost of war. The mural acts as both a celebration of resilience and a warning against the dangers of history repeating itself, urging us to remember the lessons of the past and strive for a more just and equitable future.

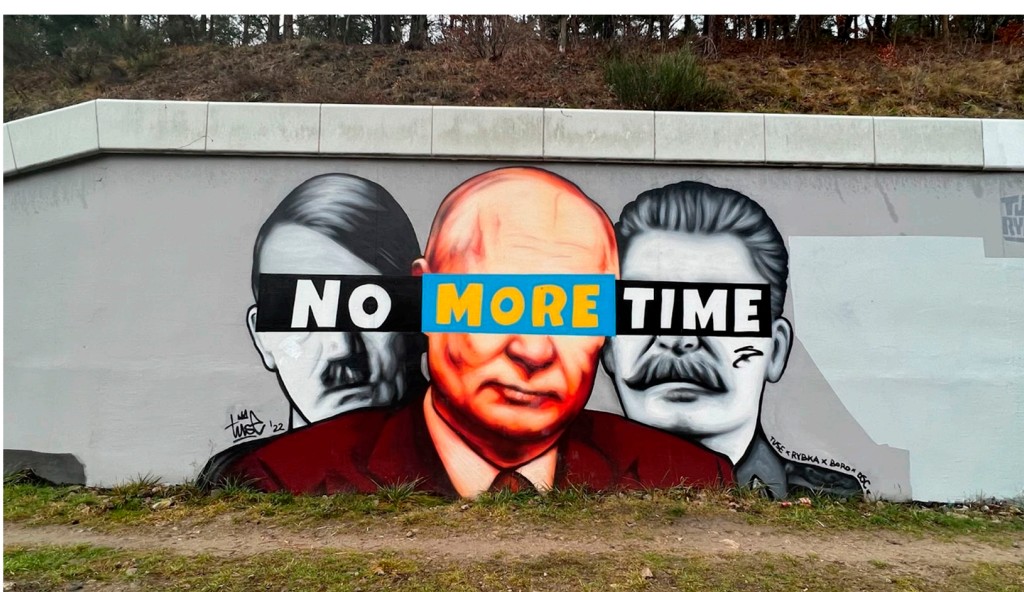

**Figure 13.** Artists: Piotr "Tuse" Jaworski, Mateusz Rybka.

The striking image of Putin alongside Hitler and Stalin serves as a chilling reminder of the dark side of power and the dangers of unchecked ambition. The obscured eyes of these war criminals, with the inscription "No More Time", is a potent metaphor for the blindness of their ideologies, and an urgent call to action against the tyranny they represent. This juxtaposition underscores the cyclical nature of history, cautioning us to remain vigilant against those who would exploit their power for nefarious ends.

The anti-war mural gallery also displays images of children struggling with fear, helplessness, loneliness, and indifference. An example is the work by Karyna Kalinka (Figure 14, which shows a lonely girl wandering with a teddy bear. The central figure of the lonely girl, clutching her teddy bear, is a haunting embodiment of innocence lost. The teddy bear, a universal symbol of childhood and comfort, serves as a poignant reminder of the stability and security that have been stripped away by the ravages of war. The girl's wandering underscores the displacement and disorientation that countless children face in times of conflict, as they are torn from their homes, families, and communities.

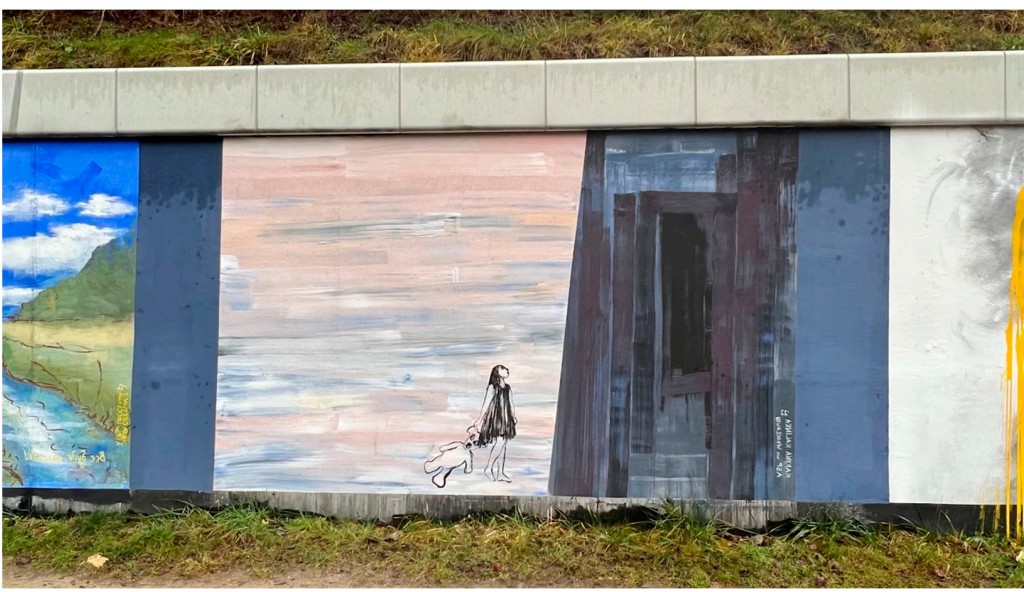

**Figure 14.** Artist: Karyna Kalinka.

The girl's vulnerability functions as a metaphor for the broader societal struggle to navigate the seemingly insurmountable chaos and devastation unleashed by war. Her solitude and isolation represent not only the deeply personal experiences of suffering, but also the collective guilt of a world that has glaringly failed to safeguard its most vulnerable constituents. The mural further delves into the issue of indifference, prodding viewers to face their own passivity in the wake of such staggering distress. By forcing viewers to bear witness to the dire predicament of the girl, the mural challenges them to recognize the devastating human toll exacted by war, and to reassess their individual roles and responsibilities in the ongoing pursuit of peace. This powerful portrayal implores viewers to contemplate the plight of the innocent, and to strive toward a world where the light-hearted laughter and carefree delight of childhood can thrive, undimmed by the looming specters of violence and despair. The abandoned child is a universal symbol of a lost childhood.

Meanwhile, Adrianna Piotrowska's mural (Figure 15) shows two children wearing the national colors of Ukraine. Their facial expressions show sadness, concern, anger, and disagreement with what is happening. These are faces of the social consequences of war. Each emotion tells a story, offering glimpses into the psychological impact of living in a war-torn landscape. Sadness speaks to the heartache of a childhood interrupted, as simple joys and carefree days are stolen away. Concern reflects the weight of uncertainties that loom large in the minds of young ones, robbed of the comforting assurances of stability and security. Anger gives voice to the visceral frustration and indignation felt by children who bear witness to the senseless violence, knowing they are powerless to stop it. Disagreement highlights the inherent injustice of war, as innocent lives are disrupted or destroyed through no fault of their own. Beyond the faces of these two children lies a much larger narrative: the story of a generation robbed of its innocence, forced to grapple with the harsh realities of life far too soon. Through the powerful imagery of Piotrowska's mural, the audience is invited to contemplate the broader social consequences of war, and to empathize with the plight of the children.

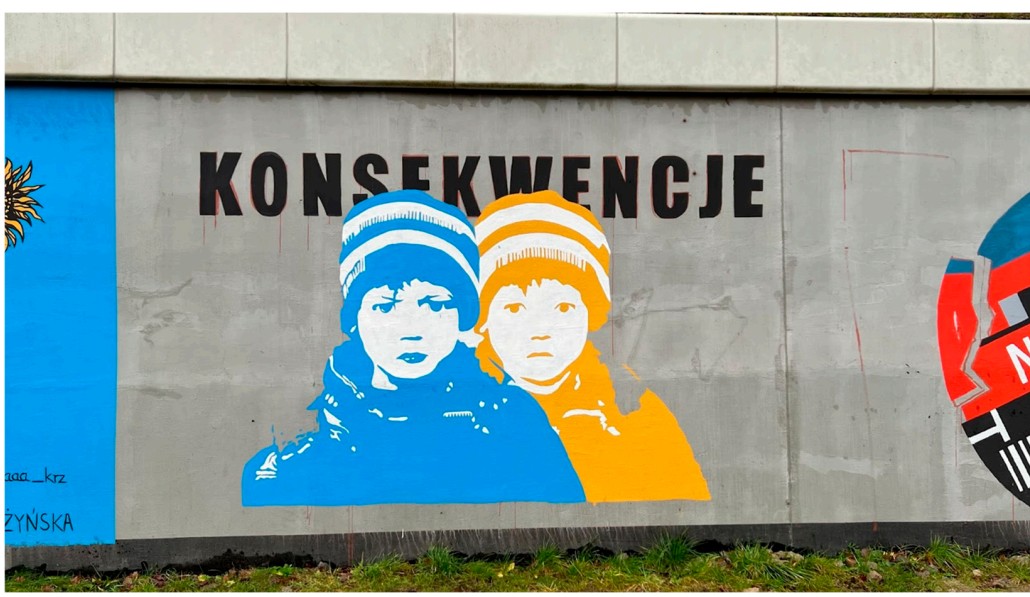

**Figure 15.** Artist: Adrianna Piotrowska.

The emotionally charged painting by Ukrainian artist Vasyl Netsko (Figure 16) speaks volumes about the harsh realities of war and the deep scars it leaves on the most vulnerable members of society. Featuring a crosshair aimed at playing children, the painting is a rifle with symbolic imagery that highlights the inherent brutality of armed conflict. The crosshair, a symbol of targeted violence, draws attention to the stark contrast between the innocence of childhood and the ruthless nature of warfare. In this metaphor, the artist illustrates how war callously encroaches on the lives of children, as if they too are mere targets on the battlefield. By capturing this juxtaposition, Netsko urges the viewer to confront the horror of violence being inflicted upon the most vulnerable. The playing children represent not only innocence and vulnerability but also the potential for growth and development. Their carefree demeanor underscores the tragedy of a childhood stolen by war, as they remain oblivious to the imminent threat that looms over them. The crosshair symbolizes the disruption of their lives, marking them as collateral damage in a conflict they neither understand nor control. Furthermore, the painting alludes to the heart-wrenching reality faced by many Ukrainian children. This cruel act serves as a metaphor for the erasure of their cultural roots, severing their connection to their heritage and traditions. As an ultimate act of cruelty and a war crime, it signifies the disregard for the sanctity of human life and the trampling of human rights.

Through his powerful artwork, Vasyl Netsko weaves a narrative that encourages viewers to empathize with the plight of children caught in the throes of war. It serves as a stark reminder of the urgent need for peace and the protection of the most vulnerable, urging society to advocate for their rights and safeguard their futures.

One of the ways artists create a powerful visual effect on murals depicting oppressors and the oppressed is using contrasting imagery. The depiction of figures such as Putin, Hitler, and Stalin alongside innocent victims, especially children, serves to highlight the huge disproportion between the two groups. This contrast not only emphasizes the enormous power wielded by these individuals but also the vulnerability and helplessness experienced by those who suffer at their hands.

The emotional impact of the murals depicting the torturers and the oppressed is an important aspect of their effectiveness. By showing the harsh realities of war and the victims' feelings of fear, helplessness, and despair, as well as anger and disagreement with the ongoing conflict, the artists are able to arouse empathy and awareness in the audience. This emotional bond encourages individuals to reflect on the consequences of war and the importance of preventing further suffering.

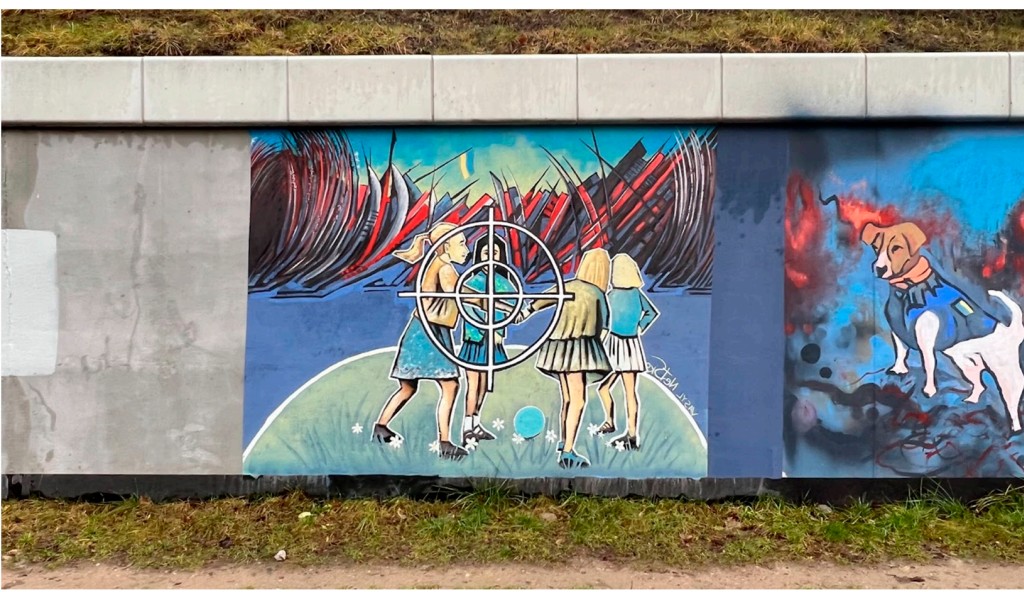

**Figure 16.** Artist: Vasyl Netsko.

Arousing empathy is central to the emotional impact of the murals, as it allows the audience to deeply connect with the depicted subjects, fostering understanding and compassion. By illustrating the torturers and the oppressed, the artists unveil the stark realities of war, highlighting not only the physical suffering but also the profound emotional and psychological toll it takes on all involved. Empathy is elicited in the audience by depicting the victims' wide range of emotions, including fear, helplessness, despair, anger, and disagreement with the ongoing conflict. This emotional resonance is crucial in fostering a sense of shared humanity and encouraging people to engage with the subject matter on a more personal level. By allowing viewers to step into the shoes of the victims and feel their anguish, the murals challenge preconceived notions and break down barriers that might hinder understanding. Moreover, empathy plays a critical role in raising awareness about the consequences of war, as it compels the audience to reflect on the cost of human lives, shattered communities, and the loss of dignity experienced by those affected. It underscores the importance of preventing further suffering, fostering peace, and promoting diplomacy as the preferred means of resolving conflicts. Empathy has the power to bridge gaps, create dialogue, and bring about transformative change by uniting people from different backgrounds and perspectives under a common cause: the pursuit of a more peaceful, just, and humane world. Therefore, the emotional connection with murals established through empathy serves as a catalyst for social change, as it motivates individuals to take action, advocate for peace, and support initiatives aimed at alleviating the pain of those suffering from war.

## 11. Murals about Animals

This thematic category encompasses vibrant depictions of birds and domesticated creatures, intended to embody and convey symbolic virtues such as unwavering loyalty, selfless aid, buoyant hope, and steadfast faith in Ukraine's ultimate triumph. One example from this category is a mural portraying a dog, a tribute to a real-life canine named Patron (Figure 17), who assists Ukrainian soldiers in their painstaking endeavors to locate and disarm mines. Set against the backdrop of a cityscape left in ruins, the canine emerges as a beacon of hope, functioning as a steadfast rescuer and defender.

In a more expansive context, the image of the dog, a deeply ingrained cultural symbol, summons a plethora of associations related to staunch loyalty, unwavering fidelity, and the instinctual readiness to extend aid (Impelluso 2006). Patron's commendable role in mine detection since the onset of the conflict has earned him the status of a local hero. His extraordinary service has been recognized by UNICEF, which bestowed upon him the

honorable title of "Goodwill Dog". His likeness now graces murals across various Ukrainian cities and adorns stamps of the Ukrainian postal service, immortalizing his heroism.

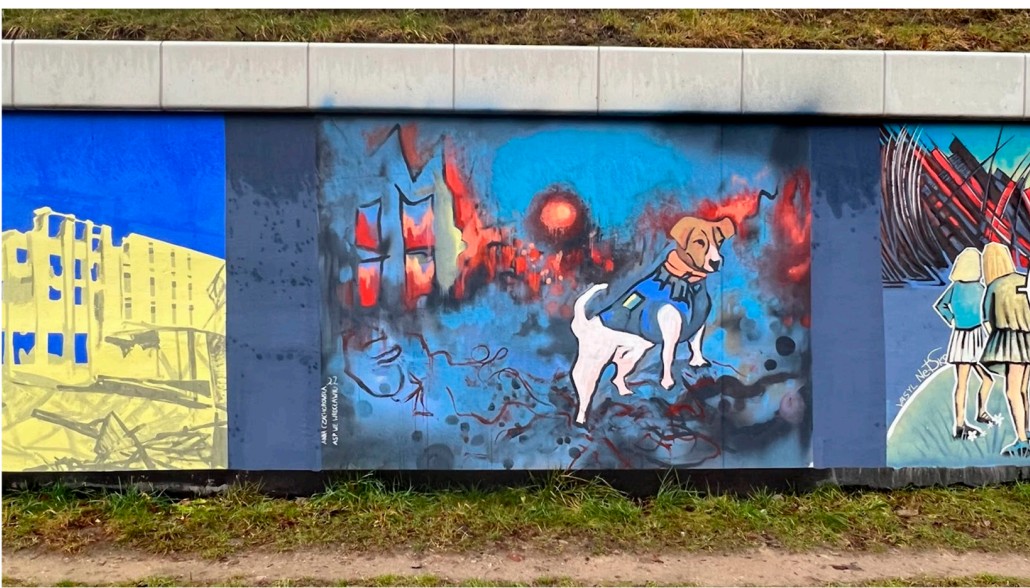

**Figure 17.** Artist: Anna Czachorowska.

Moreover, educational initiatives have been undertaken to amplify Patron's influence; he features in instructive videos providing guidance on safety protocols, further cementing his position as an emblem of hope and resilience. Thus, through the universal language of murals, Patron's narrative continues to inspire, instilling values of loyalty, courage, and relentless determination, even in the face of profound adversity.

The motif of birds echoes resonantly across the murals, their symbolic flight carrying deeply emotive messages. There are doves with olive branches, carrying a message of peace, and a nightingale that breaks through the thicket of barbed wire, symbolizing hope and perseverance in the face of constant enemy attacks (Figure 18).

The dove, cradling an olive branch in its beak, emerges as an emblem of peace, heralding the collective aspiration for the cessation of conflict and the dawn of harmony. The olive branch, an ancient icon signifying reconciliation and unity, imbues the depiction with further nuance, accentuating the profound need for dialogue, empathy, and shared understanding amidst challenging adversities.

The potent dichotomy between the dove's inherent fragility and the resilient strength of its peaceful overture underscores the transformative potential of tranquility to mend even the deepest and most scarred wounds inflicted by war. Another stirring embodiment is that of the nightingale, celebrated for its enchanting melody, which resolutely pierces through the forbidding veil of darkness and fear, cleaving a path through the tangled thicket of barbed wire that ensnares the conflict-scarred landscape.

This vivid portrayal encapsulates the unwavering spirit of hope and tenacity, signifying the bird's refusal to be subdued or constrained by the formidable physical and emotional barricades erected by war. The nightingale's melodious song, cutting through the harrowing gloom, carries with it the promise to invigorate and inspire those grappling against the overwhelming force of constant enemy assaults. It serves as a poignant reminder that, even amidst the darkest hours, the resilient spirit of beauty, hope, and endurance continues to flourish, urging us to never lose sight of the potential for a brighter tomorrow.

A tranquil scene of an idyllic landscape provides the backdrop for the diminutive yet symbolically powerful wren. Its presence within the mural is by no means fortuitous, for, in Ukraine, this little bird is interpreted as a tiny monarch, carrying the auspicious omens of joy and prosperity.

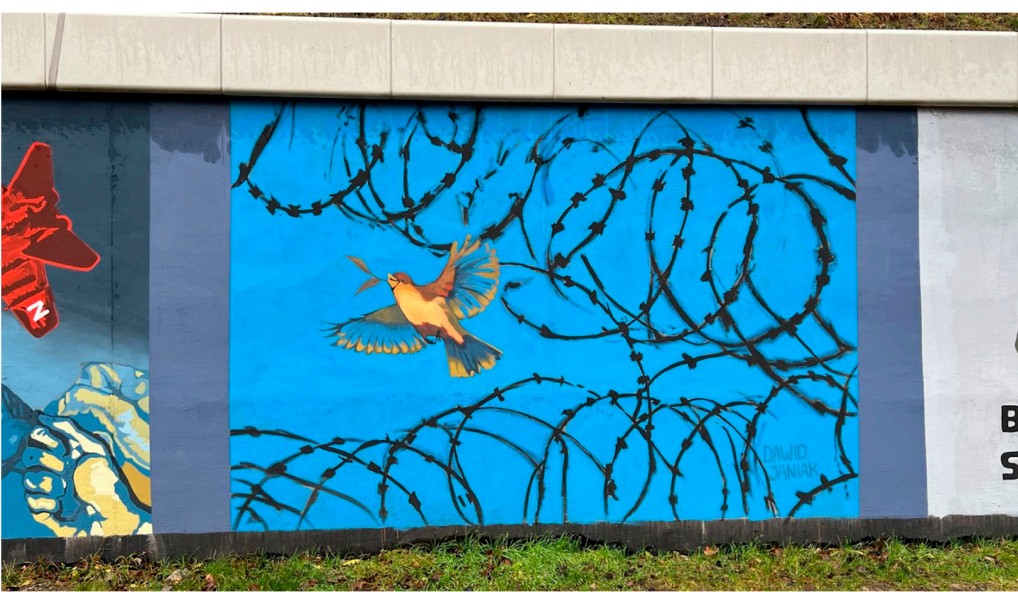

**Figure 18.** Artist: Dawid Janiak.

Interwoven with this scene is the country's national emblem, the trident or Tryzub, a potent symbol of national identity and statehood. The vibrant hues of Ukraine's national flag, blue and yellow, unfurl across the canvas, evoking imagery of the nation's azure skies and sprawling wheat fields, embodying the vibrancy and resilience of Ukraine's spirit and its undying hope for a future of peace.

These murals communicate a powerful narrative of strength in the face of adversity and the unwavering resilience of the human spirit. The bird motifs, imbued with profound symbolism, serve as beacons of light amidst the turmoil and despair of war. They inspire the observers to embrace the message of peace, hold onto hope, and persist in the struggle for a brighter and more harmonious future.

## 12. Idyllic Murals

Murals embodying pastoral tranquility feature landscapes imbued with nature's splendor and accentuated by elements of traditional Ukrainian folk art, drawing inspiration notably from the oeuvre of Maria Prymachenko. An esteemed figure in Ukrainian artistry, Prymachenko's influence resonates in the works of contemporary artists, with her distinctive style and utilization of folk motifs inspired by nature establishing her as a cornerstone of Ukraine's cultural heritage (White 2022). These murals evoke a profound tranquility, intertwining the earthly with the traditional and cultural, and serving as solemn reminders of the imperative to safeguard Ukraine's rich natural and cultural endowment. In a global context where regional values and heritages are persistently undermined and endangered, these idyllic murals resound with an emphatic call for the protection and nurturing of the unique cultural and natural riches that define Ukraine (Chmil et al. 2021).

One such example is the mural by Maja Konopacka, which employs evocative elements of folk culture (Figure 19).

The lower part of the mural (Figure 19) features embroidery, an emblem of Ukrainian folklore and a testament to the rich cultural tapestry woven by generations of artisans. Two women in folk costumes inhabit the scene, embodying the spirit and resilience of the Ukrainian people in the face of adversity. In the first perspective, a peaceful scene unfolds: the woman, surrounded by birds and clutching a musical instrument, basks in the harmony of her idyllic surroundings. The sunflower, a symbol of life, growth, and happiness, casts a warm glow on the women and the branchy tree above, representing the unity of nature and culture. At the mural's center, a blooming tree stands as a symbol of constant development, underscoring the potential for growth even amidst devastation. Yet, the tree also signifies

alternative outcomes and choices in life, emphasizing the delicate balance between life and death, or between peace and war. The falling buds, a haunting reminder of the loss of life, further accentuate this duality.

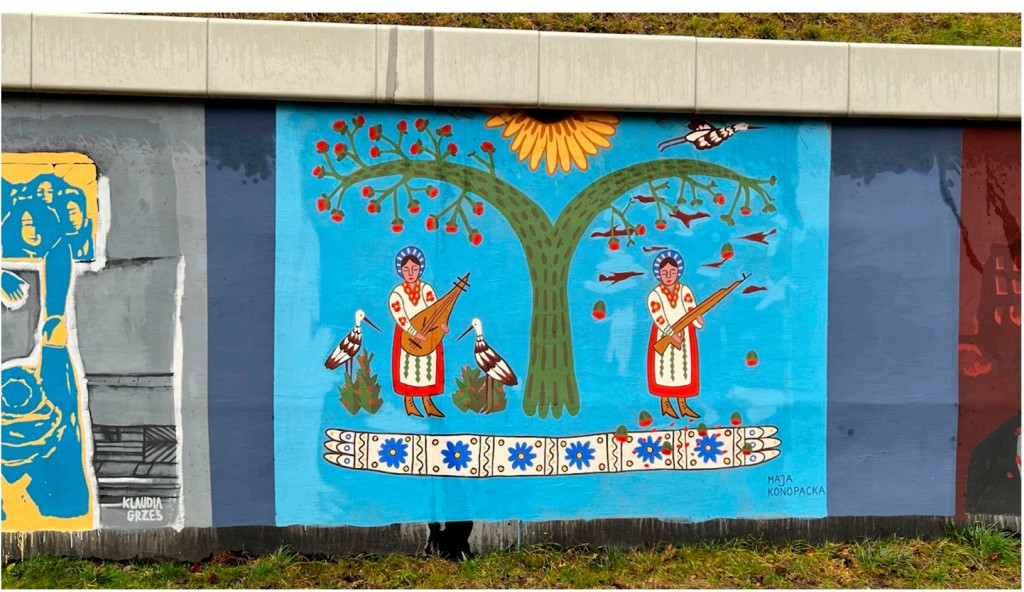

**Figure 19.** Artist: Maja Konopacka.

In stark contrast, Justyna Mężydło's mural unfurls a narrative of a woman, cradling a loaf of bread amidst a backdrop of grain fields aflame (Figure 20). Her companions, a pair of pure white pigeons, add to the evocative symbolism. The woman's traditional attire resonates with national values, while the bread she holds and the crumbs scattered on the ground echo sentiments of hope, resilience, and the intrinsic need for shared community support. In essence, drawing upon elements of folk culture, the mural vividly encapsulates the harsh juxtaposition between the serenity of peacetime and the tumultuous upheaval that comes with the outbreak of war.

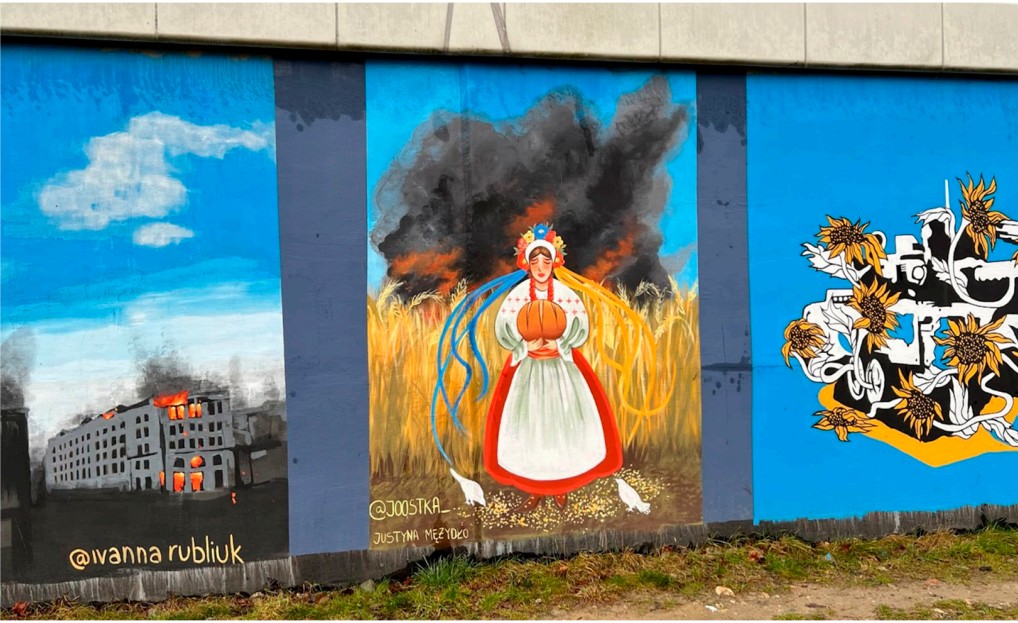

**Figure 20.** Artist: Justyna Mężydło.

The murals by Konopacka and Mężyło present a compelling invitation to their audiences. They urge contemplation on the profound transformation instigated by war and its enduring impact on cultural, personal, and collective identities, as well as the human condition as a whole. Their work echoes a poignant reminder of the delicate equilibrium between peace and conflict, and advocates for the creation of a world where the splendor of life can bloom, unburdened by the ominous specter of war.

## 13. Conclusions

Murals, embodying a myriad of contexts in which they are created, lend themselves well to a critical analysis. Our approach places emphasis on drawing correlations between historical personas and contemporaneous political events, viewed through an ethical prism. Although we did not dissect each of the 32 murals (Appendix A) that adorn the railway embankment at *Jasień* station in Gdansk, we chose emblematic examples across six categories which emerged from our content analysis, grounded in P.M. Lester's perspective.

Historically, war-related murals encourage audiences to question the status quo and consider alternative perspectives on issues related to conflict and violence. They can serve as a medium for healing and reconciliation. However, there are also people expressing opposition to this type of visual message, including in the case of murals in Gdansk, which is most often manifested by spray painting over their fragments. Notably, in July 2022, a group of teenagers left anti-Ukrainian slogans and symbols associated with the invaders on an undeveloped area of the wall (Kamasz 2022). One of the damaged murals is a work depicting the image of the Klitschko brothers, with the inscription "Together We Stand". The brothers have become local heroes, and the mural has a positive connotation. It refers to values such as honor and bravery. Both brothers—as former world heavyweight champions—could leave the country; instead, for a year, they have been actively involved in activities to defend Ukraine. It is worth noting that Witalij Kliczko (Figure 21), the mayor of Kyiv, shared photos of the murals PKM at the Gdansk *Jasień* railway station on his social media. Even more surprising is that someone damaged the mural by painting "x" over the eyes of one of the brothers. Such incidents show how extreme emotions can be evoked by anti-war murals. While some view them as symbols of resistance and societal commitment, others perceive them as propagandist instruments.

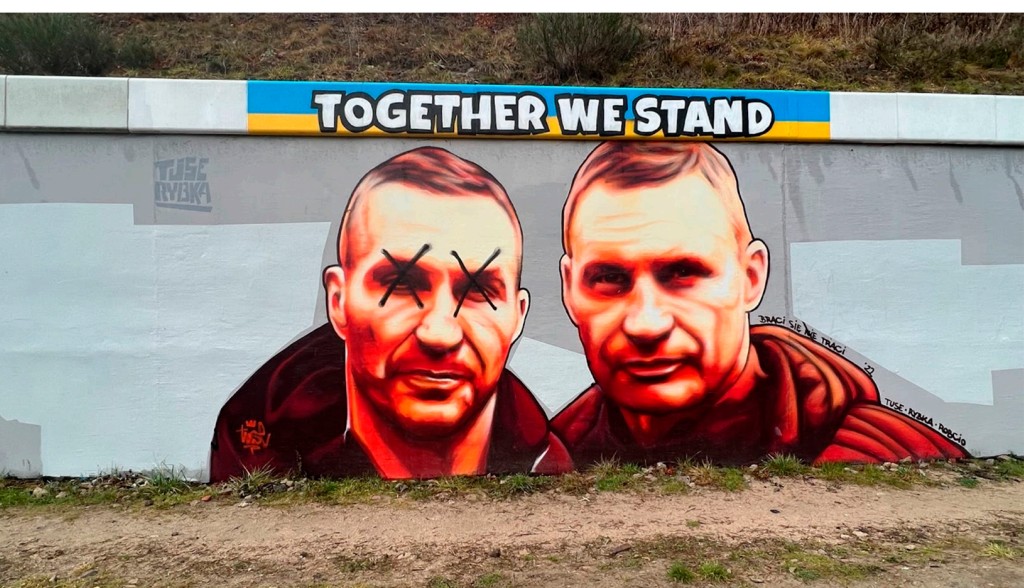

**Figure 21.** Artists: Piotr "Tuse" Jaworski and Mateusz Rybka.

The personal perspective of the authors of the murals is significant. They were looking for a suitable form to express their values. The authors of the murals in Gdansk are students coming not only from Poland but also from Ukraine and Belarus, which translates into

various forms and symbols used. It also shows that street art has a wide range of social impacts, and that its effects depend on the views, knowledge, and competencies of not only the creators themselves but also the audience.

Although this gallery of anti-war murals contains many sad and moving images, each one carries a message of hope. It communicates the need to recall what is essential for victory, referring to the allegory of the spirit of freedom and steadfastness. The content of the murals evokes extreme emotions—from fear and doubt to courage and faith in the victory of those who fight for the good of their own country.

Anti-war murals perform essential social and educational functions. They speak out on important matters, informing with their message about what should never be passed over in silence. By illustrating the human costs of armed conflict, the murals raise awareness of the plight of refugees, the impact on families and communities, and the importance of humanitarian aid. This heightened impact on consciousness can inspire people of all ages to take action and engage in initiatives to support the war-affected population.

An essential aspect of murals is their accessibility. The landscape of urban space, often saturated with commercial advertising media of dubious aesthetic value, gains a more profound meaning thanks to works of art. These works do not serve commerce or drive consumerism; they are a voice in socially essential matters. This is achieved by carefully selected visual elements, such as colors, shapes, and figures, evoking emotions of fear, anger, rage, and sadness. With their calmness and intensity, they inform about what should never be left unsaid and add faith that this image will no longer be needed one day. Murals convey complex ideas and emotions in a way that is both immediate and evocative. They have local potential that can be worth expanding in the case of murals in *Jasień*.

Photos of some of the murals and short video materials are available on the fan page of the Metropolitan Railway on the Internet. However, what is missing is an online gallery containing photos of all the works created so far, which would make them more accessible to the public. In addition, the creation of such a gallery would have many benefits.

Amplified awareness: By consolidating murals into one readily accessible digital space, the gallery amplifies the art form's visibility, fostering broader public cognizance and esteem for the mural art's capacity to communicate and inspire.

Inclusive documentation: A unified online platform harboring all the murals ensures that no works slip under the radar, offering a more encompassing portrayal of the art form and its thematic diversity.

Expanded accessibility: An online gallery allows a global audience to observe and appreciate murals unimpeded by geographical constraints, broadening the artists' reach and the influence of their messages.

Promoting a peaceful culture: The gallery encourages a culture of peace and reverence for other nations, bolstering public consciousness of the significant societal impact of mural art.

Highlighting regional and national values: The gallery becomes an instrument to safeguard cultural heritage for future generations, underscoring national and regional values, and increasing public awareness of the importance and social impact of mural art.

Rethinking urban space: It triggers contemplation on the urban space as a canvas enabling the creation of legal, large-format artistic works.

The socio-educational potential of the *Jasień* gallery in Gdansk does not differ significantly from what inspired its creators in the past (Pawłowska et al. 2023). Its murals increase social awareness while being an exciting and valuable combination of aesthetics and involvement in social issues. As part of local activities, they can integrate and become a contribution to conversations that—perhaps—would not take place in other circumstances.

**Author Contributions:** Conceptualization, E.P.-B.; formal analysis, M.G.; resources, M.P. and K.K.; data curation, M.G.; writing—original draft, K.K.; supervision, E.P.-B. All authors have read and agreed to the published version of the manuscript.

**Funding:** This research received no external funding.

**Data Availability Statement:** Not applicable.

**Conflicts of Interest:** The authors declare no conflict of interest.

## Appendix A

**Table A1.** A list of 32 murals at the Jasień Railway Station in Gdansk. The authors do not give the murals' titles. Each mural is a rectangle with the dimensions listed.

| No. | Name and Surname of the Author of the Murals | Photo of the Mural | Description of the Mural from the Authors' Statements | The Dimensions |
|---|---|---|---|---|
| 1. | Piotr "Tsue" Jaworski |  | The mural depicts the likeness of the leader of the USSR, Joseph Stalin, the leader of the Third Reich, Adolf Hitler, and the current president of Russia, Vladimir Putin. The mural is accompanied by the slogan "No More Time". | height: 2.5 m width: 3.5 m |
| 2. | Piotr "Tsue" Jaworski |  | From the beginning of the Russian invasion of Ukraine, the brothers joined the preparations for the oberona of Kiev; the elder, Vitaly, is the mayor of the Ukrainian capital. The mural is accompanied by the slogan "Together We Stand". | height: 2.8 m width: 3.5 m |
| 3. | Karol Filip Madyjewicz |  | It took place before the outbreak of the war, so the theme was free, and the mural itself was dedicated to a woman named Marcelina. On the official funpage of the Pomeranian Metropolitan Railway, it is called "An Eye for the Railway". | height:2.5 m width: 3.5 m |
| 4. | LUSSO ART. ATEUER Nina Sobczak |  | The mural I painted on the wall for Jasień PKM was the second one that appeared there before the outbreak of the war in Ukraine. The inspiration for painting was one of my favorite motifs—forest and owl. The idea behind the composition was to frame the forest scene. It's like we've frozen a frame from a movie. An interesting fact is that, when looking at this mural at night, only the owl itself is visible. | height: 2.5 m width: 3.5 m |
| 5. | OKŁ P. Wroniszewski |  | The mural with the image of a cow refers directly to the music album entitled "Masakra", released in 1998 by the rock band "Republika". The logo with a cow is a faithful representation of the graphics on the cover of the studio album. | height: 2.5 m width: 2.2 m |
| 6. | Paulina Sosińska |  | The mural symbolizes a sudden attack by Russian forces on the Ukrainian nation, breaking all the rules of war. It is a symbol of support for Ukraine and a warning that, even in today's world, we cannot feel safe. | height: 2.5 m width: 3.5 m |
| 7. | Ivana Rubliuk |  | Title: Ukraine Is a Woman. The mural depicts the mother of Ukraine, who, together with heroic soldiers, will not allow our Ukraine to be annihilated. The strength does not lie in the modern armament of the enemy, but in the spirit, devotion, and faith of the people of Ukraine. Glory to Ukraine! Many thanks to Poland! | height: 2.5 m width: 3.5 m |

**Table A1.** *Cont.*

| No. | Name and Surname of the Author of the Murals | Photo of the Mural | Description of the Mural from the Authors' Statements | The Dimensions |
|---|---|---|---|---|
| 8. | Justyna Mężyło |  | The impulse to create the murals was the bombing of the museum by the Russians with the works of the Ukrainian artist and peasant Maria Prymachenko. I referred to one of her paintings, which shows a Ukrainian woman sharing bread. However, in my project, a woman sharing bread is shown against the background of burning grain fields, destroyed by the war. The color of the blue, burning sky and grain refers to the Ukrainian flag, the figure of a woman symbolizes the suffering of people at war, and the bread symbolizes hope. | height: 2.5 m width: 2.2 m |
| 9. | Maja Krzyżyńska |  | The Russian T-90 Vladimir tank was sent to the front as part of the invasion of Ukraine. One shot was enough for the machine proudly advertised by the Russians to be defeated. The sunflowers growing on the tank symbolize the freedom and independence of Ukraine. I designed my mural to support visiting families and to express my disagreement with the current situation in an artistic way. | height: 2.5 m width: 3.5 m |
| 10. | Adrianna Piotrowska |  | The message of the mural is direct, referring to one of the many consequences after the war. It draws attention to children affected by the war. They will remember these horrific images for a long time. Different emotions are drawn on the faces of the two: on one, there is sadness and confusion, and, on the other, anger is drawn. | height: 2.5 m width: 3.5 m |
| 11. | Iwona Sidło |  | The "No Signal" mural is a reaction to what is happening beyond our eastern border. He breaks away from strictly war topics, pointing to the lack of contact between Moscow and the West. Its meaning is both symbolic and literal. It symbolically emphasizes the narrow-gauge, outdated way of thinking, detached from that flowing in the Western world. It literally indicates the absence of a signal or the presence and free flow of information. | height: 2.5 m width: 3.5 m |
| 12. | Natalia Świerczyńska |  | The mural was inspired by a unique photograph that circulated around the world at the beginning of March. It shows a woman walking with a child and carrying an AK-74 automatic carbine on her back. The photos taken in the first days of the war at the main station in Kiev symbolize the courage and commitment of the Ukrainians in the face of the illegal Russian invasion. | height: 2.5 m width: 3 m |
| 13. | Maria Morawska |  | Firefighters around the world, every day, do what they can to save the earth, buildings, and lives of others from the element that destroys. I want my mural to be a kind of commemoration of what they had to face. At the same time, I want to pay tribute to their work, strength, dedication, and heroism that they show every day. | height: 2.5 m width: 3.9 m |

**Table A1.** *Cont.*

| No. | Name and Surname of the Author of the Murals | Photo of the Mural | Description of the Mural from the Authors' Statements | The Dimensions |
|---|---|---|---|---|
| 14. | Denis Chyżawski |  | The mural shows a symbolic Ukrainian field of wheat, the ears intertwine in such a way that they form the silhouette of the Ukrainian weapon Malik/Volcano, giving hope to the entire Ukrainian Nation for victory, at the same time showing the strength and power of Ukraine. | height: 2.5 m width: 3.2 m |
| 15. | Milana Chyżawska |  | Russia started a war in Ukraine. In many cities and towns of Ukraine, Russian soldiers are killing civilians. The mural shows the horror of the war against the Ukrainian people in 2022. | height: 2.5 m; width: 1.8 m |
| 16. | Agnieszka Lewandowska |  | A fighter marked with the letter "Z" is attacking the fertile fields of Ukraine, confusingly resembling a red star, the symbol of the Soviet army. Joined in solidarity in one blow, the blue fists repel the enemy's attack. It is not known what will happen next, but the bottom line is certain—this is a decisive moment in the history of the world. | height: 2.5 m width: 3.2 m |
| 17. | Dawid Janiak |  | The mural depicts a nightingale, the national bird of Ukraine, flying through a tight thicket of barbed wire. In its beak, it holds a fragment of an olive branch, widely associated with peace. Painted mainly in the colors of the Ukrainian flag, both the bird and the blue sky shining through the wire are meant to be a symbol of hope and perseverance in the face of an attack. | height: 2.5 m width: 3.6 m |
| 18. | Natasza Goliszek |  | In the first weeks of the war, I came across news about Lesio and Valeria's wedding. The couple decided that the war would not be an obstacle to the wedding. They got married at the front near Kiev. The mural gave hope for a better tomorrow, despite the tragic circumstances. | height: 2.5 m width: 3.3 m |
| 19. | Klaudia Grześ |  | The work was inspired by personal observation of the consequences of events affecting the Ukrainian population. | height: 2.5 m; width: 1.8 m |
| 20. | Maja Konopacka |  | I was inspired to create the mural by Ukrainian folk painting, mainly the works of Maria Prmaszenko, some of which were burnt down during Russian attacks aimed at destroying Ukrainian culture as well. In a symbolic way, I present the changes that the war has forced, both in people and in the environment. | height: 2.5 m width: 3 m |
| 21. | Natalia Jaskólska |  | The mural depicts a rabid, aggressive dog that demolished the city and a punishing, lecturing hand. The illustration symbolizes direct and inhuman aggression against Ukraine, which, despite the destruction and losses, tries to repel, warn, and educate its occupiers. | height: 2.5 m width: 3 m |

**Table A1.** *Cont.*

| No. | Name and Surname of the Author of the Murals | Photo of the Mural | Description of the Mural from the Authors' Statements | The Dimensions |
|---|---|---|---|---|
| 22. | Katarzyna Witek |  | I was inspired to create the mural by Ukrainian embroidery, which is an important element of the culture and national identity of Ukraine. The theme of the work is a symbolic representation of alienation, helplessness, and loneliness in the face of the war in Ukraine, which affects many of us. | height: 2.5 m width: 3.3 m |
| 23. | Aliaksandr Danilkin |  | Hakenkreuz = Z. | height: 2.5 m width: 3.1 m |
| 24. | Zuzanna Kłapkowska |  | The folk art of the Eastern European culture is, in a certain way, coherent. Common motifs, colors, and shapes interpenetrate. It is a universal language that can be used to describe various states and empathy. This form of communication gives the opportunity to experience current events together. | height: 2.5 m width: 3 m |
| 25. | Olga Kossmann |  | The composition shows the outline of shadows emerging from a fire symbolizing war. The characters break through the bloody darkness, uniting by saying "stop the fight" through a familiar gesture. The hands were painted in the colors of the Ukrainian flag, and the red refers to the aggressor—Russia. | height: 2.5 m width: 4.5 m |
| 26. | Prof. Adam Chmielowiec & asystent Tomasz Wiktor |  | A scratch—a frozen trace, definitely lost in matter, not so much written as permanently imprinted in our self. A commentary on the experience in the subjective manifestation of freedom. A sign, hopefully temporary, soon just a memory. | height: 2.5 m width: 3 m |
| 27. | Karyna Kalinka |  | The human experience of war, the sense of hopelessness, the loss of childhood. Helplessness against something that is many times bigger and stronger than you. | height: 2.5 m width: 3.5 m |
| 28. | Julia Grzywacz |  | The project presents the idyllic landscape of the Ukrainian countryside in the summer—ears of corn, haystacks, a roaring, cool river, and a forest looming in the distance. | height: 2.5 m width: 3.3 m |
| 29. | Aleksandra Kamińska |  | Carefree healing birds holding olive branches in their beaks—a symbol of peace in the world, ubiquitous in culture. Birds flying over the territory of Ukraine are dreams that, hopefully, will come true any moment. | height: 2.5 m width: 3.3 m |

**Table A1.** *Cont.*

| No. | Name and Surname of the Author of the Murals | Photo of the Mural | Description of the Mural from the Authors' Statements | The Dimensions |
|---|---|---|---|---|
| 30. | Julia Kochańska |  | Will—that is, the desire to achieve the set goal despite all obstacles. It perfectly describes the strength of the Ukrainian nation. With my project, I want to show that, despite the enormity of victims and all the hardships that Ukraine is suffering, it still has strength and support that will ensure its freedom. | height: 2.5 m width: 3.3 m |
| 31. | Anna Czachorowska |  | In my project, I wanted to approach the subject in a visual way. I placed the ruins and the dog Patron on it, helping Ukrainian soldiers in search of mines. | height: 2.5 m width: 3.3 m |
| 32. | Vasyl Netsko |  | A crosshair aimed at playing children says it all in my opinion. Children are what is most important and what is most dear to a person; they are the future. | height: 2.5 m width: 3.3 m |

Source: Solidarity with Ukraine. The largest gallery of anti-war murals in Pomerania at the Gdansk *Jasień* railway station: www.pkm-sa.pl (accessed on 14 April 2023).

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
