# Peer review of "Socio-Educational Impact of Ukraine War Murals: Jasień Railway Station Gallery"

_arts, 2023_

Round 1

Reviewer 1 Report

It would be reasonable to reinforce the Ukrainian perspective on the perception of the war through murals, e.g. the article: 

PawÅ‚owska Aneta, GraliÅ„ska-Toborek Agnieszka, at al.,   Problems of Expositions and Protection of Banksy’s Murals in Ukraine, International Journal of Conservation Science, 2023, vol. 14, no 1, pp.99-114

  Publications should also be taken into account:   Agnieszka GraliÅ„ska-Toborek " Grafitti i street art. SÅ‚owo, obraz, dziaÅ‚anie" (due to the selection of the bibliography the author is a Polish scholar so the language of the publication is not a problem).

The language is correct but not too rich.

Author Response

Dear Reviewer,

We would like to express our sincere gratitude for your insightful comments and constructive criticisms of our manuscript entitled "Socio-Educational Impact of Ukraine War Murals: Jasień Rail-way Station Gallery." Your feedback was essential for improving the quality of our paper.

Firstly, we appreciate the acknowledgement of the minor English language edits required. We have thoroughly revised the manuscript for grammatical and typographical errors. Our efforts were focused on enhancing the readability of the text, ensuring that the language is clear, precise, and rich. As suggested, we have enriched the descriptions of the murals, providing more detailed and vibrant descriptions to evoke a more substantial impression for the reader. We trust that this more descriptive language enhances the understanding and appreciation of the murals and their socio-educational impact.

Secondly, we are also to reinforce the Ukrainian perspective on the perception of the war through murals. This includes integrating two recommended publications into our analysis, which have now been properly cited. These additional sources have not only enriched our discussion but have also strengthened our arguments.

We hope that the revised manuscript now meets your requirements, and we look forward to your further comments, if any. We remain very grateful for your input, which is instrumental in improving the quality of our manuscript.

Thank you once again for your time and insightful feedback.

Best regards,

Authors

Reviewer 2 Report

Pertinent theme. Good structure and updated theoretical basis. Well structured exposition of the theme, purpose and objectives. Glabally good in what concerns the formal structure and thread of the research.    

It is suggested the creation of a table (summary table) with the name of the artistic works, the name of the author of the works and their dimensions.   

Author Response

Dear Reviewer,

We are writing to thank you for the thoughtful and constructive feedback on our manuscript, "Socio-Educational Impact of Ukraine War Murals: Jasień Rail-way Station Gallery". Your insights and comments were immensely helpful and have guided our revisions.

We appreciate your indication of the requirement for minor edits in the English language used in the manuscript. Your comments on the language used in describing the murals in the reporting part and the summary were well noted. We have revised these sections and enriched the descriptions. Our aim has been to create a vivid and engaging account that accurately reflects the nuances and significance of the murals.

In response to your suggestion to create summary compiled comprehensive table for all 32 murals in the order of their occurrence, which can be found in Appendix A. The table provides the name and surname of the mural's author, a thumbnail of the photo of the mural, a description derived from the statements of the mural authors, and the dimensions of the mural. We believe that this additional resource will enhance the reader's understanding of the scope and details of the mural collection.

We sincerely hope that these revisions adequately address your comments and suggestions.

Once again, thank you for your time and efforts in reviewing our work.

Best regards,

Authors
